# Scaffold-hopping for molecular glues targeting the 14-3-3/ERα complex

Markella Konstantinidou [1] ✉, Marios Zingiridis [2], Marloes A. M. Pennings [3], Michael Fragkiadakis[2], Johanna M. Virta[1], Jezrael L. Revalde [1], Emira J. Visser[3], Christian Ottmann [3], Luc Brunsveld [3], Constantinos G. Neochoritis [2] ✉ & Michelle R. Arkin [1] ✉

Molecular glues, small molecules that bind cooperatively at a protein-protein interface, have emerged as powerful modalities for the modulation of protein-protein interactions (PPIs) and "undruggable" targets. The systematic identification of new chemical matter with a molecular glue mechanism of action remains a significant challenge in drug discovery. Here, we present a scaffold hopping approach, using as a starting point our previously developed molecular glues for the native 14-3-3/estrogen receptor alpha (ERα) complex. The novel, computationally designed scaffold is based on the Groebke-Blackburn-Bienaymé multi-component reaction (MCR), leading to drug-like analogs with multiple points of variation, thus enabling the rapid derivatization and optimization of the scaffold. Structure-activity relationships (SAR) are developed using orthogonal biophysical assays, such as intact mass spectrometry, TR-FRET and SPR. Rational structure-guided optimization is facilitated by multiple crystal structures of ternary complexes with the glues, 14-3-3 and phosphopeptides mimicking the highly disordered C-terminus of ERα. Cellular stabilization of 14-3-3/ERα for the most potent analogs is confirmed using a Nano-BRET assay with full-length proteins in live cells. Our approach highlights the potential of MCR chemistry, combined with scaffold hopping, to drive the development and optimization of unprecedented molecular glue scaffolds.

The stabilization of native protein-protein interactions (PPIs) with small molecules offers an attractive strategy for the activation or inhibition of signaling pathways in a therapeutic context[1,2]. PPIs were traditionally considered difficult targets due to the lack of well-defined pockets and the presence of large, hydrophobic surfaces[3–5]. The fundamental understanding of the mechanism of action of molecular glues – small molecules that bind cooperatively at PPI interfaces and strengthen weak, pre-existing interactions – has enabled the stabilization of PPIs by taking into account the elements of cooperativity, molecular recognition and shape complementarity[6,7].

A particularly challenging class of PPIs includes proteins that are intrinsically disordered and only become partially structured when bound to a protein partner, such as a chaperone binding to a client protein[8,9]. 14-3-3 is an abundant scaffolding protein that recognizes specific phospho-serine or phospho-threonine motifs on disordered domains of the client and upon binding creates a structured binding interface[10]. Of note, 14-3-3 proteins lack function – the function is instead encoded on the client protein and in particular on the phospho-site that is being recognized, leading either to activation or inhibition of signaling pathways[11].

[1]Department of Pharmaceutical Chemistry and Small Molecule Discovery Centre (SMDC) University of California San Francisco (UCSF), San Francisco, CA, USA. [2]Department of Chemistry University of Crete Voutes, Heraklion, Greece. [3]Laboratory of Chemical Biology, Department of Biomedical Engineering and Institute for Complex Molecular Systems (ICMS) Eindhoven University of Technology, 5600 MB Eindhoven, The Netherlands. ✉e-mail: markella.konstantinidou@ucsf.edu; kneochor@uoc.gr; michelle.arkin@ucsf.edu

Among the extensive interactome of 14-3-3, here we focus on its native interaction with the hormone regulated transcription factor estrogen receptor α (ERα). 14-3-3 recognizes the protein sequence surrounding phospho-T594 on the disordered C-terminus on the F-domain of ERα and acts as a negative regulator by inhibiting ERα binding to chromatin and blocking ERα-mediated transcription[12,13]. To date, ERα small molecule drugs, acting either as inhibitors or degraders, target the adjacent ligand binding domain (LBD)[14–17]. However, mutations in the LBD are often associated with acquired endocrine resistance[18]. Thus, stabilization of the native 14-3-3/ERα PPI could be useful as an alternative strategy to block ERα transcriptional activity in ERα positive breast cancer, especially in cases of acquired endocrine resistance. The feasibility of this approach, targeting the F-domain, is corroborated by studies using the natural product fusicoccin-A (FC-A) and its semi-synthetic analogs that stabilize the interaction between 14-3-3 and the C-terminus of ERα[19]. Drug-like chemical probes and leads are now required to define the biological impact of targeting the F-domain to inhibit ERα in hormone-positive breast cancer.

We have applied different strategies for the identification of chemical matter to stabilize the 14-3-3/ERα complex. We used a site-directed fragment-based technology, termed "disulfide tethering" with intact mass spectrometry as the readout to identify reversible fragments bound at the native cysteine (C38) of 14-3-3σ in the presence of a phosphorylated peptide representing the disordered C-terminus of ERα[20]. Rational, structure-guided optimization of the reversible disulfide fragments led to irreversibly covalent, selective molecular glues that bound at the composite surface of 14-3-3σ/ERα[21]. For the development of non-covalent molecular glues, we used a fragment-linking approach, derived from the crystal structures of two diverse fragments that were identified in crystallographic and disulfide tethering screens[22]. Thus, the 14-3-3/ERα PPI has served as a valuable system to test diverse molecular-glue discovery strategies.

Here, we present a scaffold-hopping approach based on multi-component reaction chemistry (MCR). Multi-component reactions are defined as synthetic approaches where at least three starting materials react in a single step to form a complex scaffold, where essentially all or most of the atoms contribute to the newly formed product. This broad definition covers reactions with various synthetic mechanisms[23,24]. MCR chemistry, due to its highly divergent character, is an enticing strategy for developing new scaffolds and rapid structure-activity relationships (SAR), as it allows the combination of short synthetic routes with high diversity and complexity. MCR has emerged as an attractive alternative to multistep linear convergent synthetic approaches and has been successfully applied to the synthesis of active pharmaceutical ingredients (API)[25–30]. Here, we describe our strategy for the development of a drug-like MCR scaffold stabilizing the 14-3-3σ/ERα complex. The most potent analogs of the series showed efficacy in orthogonal biophysical assays and cell-based PPI stabilization in the low micromolar range.

## Results

### Structure activity relationships (SAR)

Structurally, 14-3-3 binds ERα by recognizing phospho-T594, the penultimate residue on the C-terminus of ERα. This creates a large, open, solvent-exposed pocket that can accommodate a small molecule. Although steric factors are not an issue for targeting the composite surface of the 14-3-3/ERα complex, we found it was important to rigidify an initially flexible scaffold to maximize the stabilization effect[21]. Our aim in this work was to design a scaffold that would be more rigid from the beginning, locking in a favorable three-dimensional shape complementary to the large pocket.

To this end, we used the freely accessible software AnchorQuery™, which performs pharmacophore-based screening of approximately 31 million compounds that are readily synthesizable through one-step multi-component reaction (MCR) chemistry[31,32].

Although AnchorQuery™ was originally developed for PPI inhibitors[33], in this case, it was successful in proposing MCR scaffolds for PPI stabilizers. AnchorQuery™ requires a ligand-bound crystal structure or docked binding pose as a starting point. We used the crystal structure of the previously disclosed compound 127 (PDB 8ALW) that was bound at the composite surface of the 14-3-3σ/ERα complex, with a favorable ligand conformation, based on our biophysical data[21]. The compound formed multiple favorable interactions both with 14-3-3σ and the phospho-ERα peptide (Fig. 1a). In the co-crystal structure, the irreversible chloroacetamide warhead of compound 127 formed a covalent bond with C38 of 14-3-3σ. The p-chloro-phenyl ring occupied a small hydrophobic pocket that formed a halogen bond with K122 of 14-3-3. The tetrahydropyrane ring adopted a favorable conformation that allowed the formation of hydrophobic interactions with 14-3-3 residues (L218, I219), the terminal Val595 of ERα, and a water-mediated hydrogen bond from the oxygen atom in the ring, which was part of a large water network. The overall ligand conformation also led to a key water-mediated hydrogen bond between the aniline nitrogen and the terminal carboxylic acid of Val595 of ERα, significantly contributing to molecular recognition. In this compound series, the introduction of large non-aromatic rings, such as the tetrahydropyrane, combined with aniline rings, was necessary to limit the multiple ligand conformations and improve the affinity to the complex.

Understanding the binding mode of 127, we were able to use AnchorQuery™ for a scaffold-hopping approach. The software contains a virtual library; to query the library, the user defines two feature sets that derive from the binding mode of the initial ligand (compound 127, in this case). The first feature is an anchor motif on the ligand that is bioisosteric with an amino acid residue. A suitable anchor in our case is the p-chloro-phenyl ring that is deeply buried at the PPI interface in the small pocket surrounding K122 of 14-3-3 (Supplementary Fig. 1). This motif serves as the "phenylalanine anchor", which is kept constant for all pharmacophore-based searches of the database. The second feature set consists of three additional pharmacophore points on different parts of the ligand; the three-point pharmacophore could include all possible ligand-protein interactions, such as hydrogen bond donors or acceptors, aromatic rings, hydrophobic residues, or charged ions. Testing different combinations of pharmacophore points allows greater chemical space coverage and the rapid optimization of ligand-protein interactions that might be sub-optimal in the original ligand. We apply an additional filter, limiting the molecular weight of the hits to 400 Da. AnchorQuery then screens a 31 M+ member library (2 B conformers) from 27 MCR reactions, covering a large chemical space of different MCR scaffolds (from typical Ugi reactions to heterocyclic scaffolds, such as tetrazoles, thiophenes, imidazo[1,2-a]pyridines, pyrroles, and so on).

Based on the selected pharmacophore points, the software provided a list of potential docking hits from all possible scaffolds ranked according to energy minimization or RMSD fit. Since we were aiming for hits with a similar 3D shape to compound 127, we used the RMSD fit for hit ranking. We screened all 27 MCR reactions, but interestingly, all our top hits were based on the Groebke-Blackburn-Bienaymé (GBB) three-component reaction (GBB-3CR)[34–36]. The GBB-3CR utilized aldehydes, 2-aminopyridines, and isocyanides, leading to imidazo[1,2-a]pyridines[37,38]. This privileged scaffold has been present in several clinical candidates and marketed drugs, such as zolpidem, miroprofen, minodronic acid, and olprinone[39].

Comparison of the docking pose of the proposed GBB compounds with the co-crystal structure of compound 127 revealed a considerable shape complementarity and an almost identical 3D shape of the two scaffolds (Fig. 1b). Additionally, the GBB scaffold was drug-like and more rigid compared to our original ligand, potentially restricting the possible ligand conformations. After selecting the main GBB scaffold, we performed more rounds of docking and selection of suitable substituents to achieve favorable ligand–protein interactions

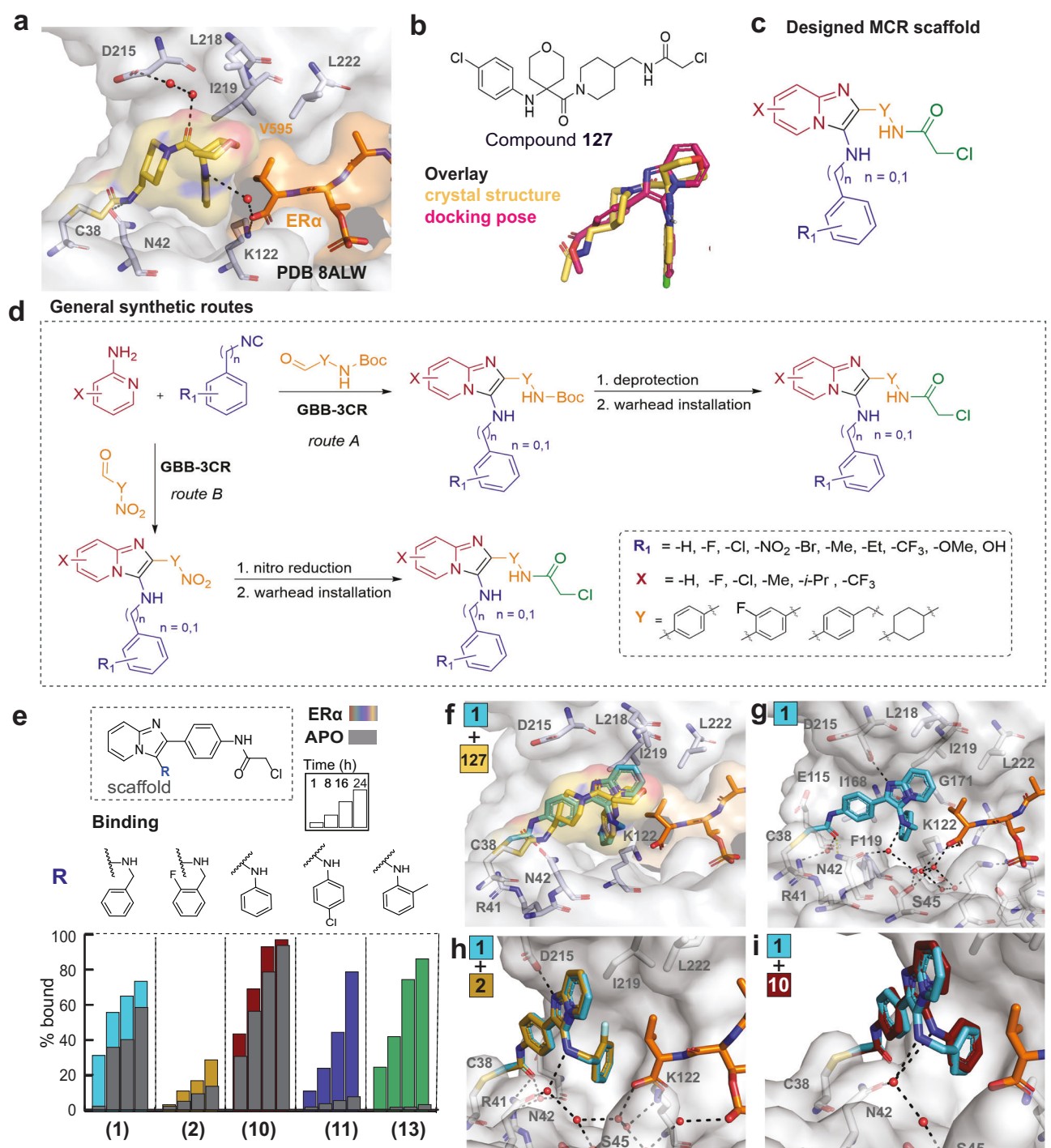

**Fig. 1 | Overview of the scaffold hopping approach. SAR and crystal structures of selected benzyl and aniline analogs. a** Crystal structure of compound **127** (yellow sticks) with 14-3-3σ (grey surface) and phospho-ERα peptide (orange sticks). Interacting aminoacids are shown as sticks and water molecules as red spheres (PDB 8ALW). **b** Chemical structure of compound **127** and ligand overlay of compound **127** and docking pose of the new MCR scaffold. **c** General multi-component reaction scaffold (MCR) based on the Groebke-Blackburn-Bienaymé (GBB) reaction and main points of variation. **d** Overview of general synthetic routes. Detailed experimental conditions are described in the SI. **e** Mass Spectrometry (MS) bar graphs at 1 μM. For each compound, time course experiments were performed with measurements at 1 h, 8 h, 16 h and 24 h. ERα data are shown with different colors for each compound, and apo data in grey. **f** Crystal structure overlay for compounds **1** (cyan sticks) (PDB 9I6Y) and **127** (yellow sticks) (PDB 8ALW) bound to 14-3-3σ (grey surface) /ERα (orange sticks). **g** Crystal structure of compound **1** (cyan sticks) (PDB 9I6Y) with 14-3-3σ/ERα. Interacting aminoacid residues are shown as sticks and interacting water molecules as red spheres. **h**, **i** Structural overlays of compounds **2** (brown sticks) (PDB 9I6Z), and **10** (dark red sticks) (PDB 9I72) with compound **1** (cyan sticks) (PDB 9I6Y).

with the docking software SeeSAR [version 14.0.0; BioSolveIT GmbH, Sankt Augustin, Germany, 2024, www.biosolveit.de/SeeSAR]. For molecular glues, the goal of structure-guided design was to incorporate the optimal ligand interactions in the composite surface, thus simultaneously improving the interactions with 14-3-3 and ERα. Although MCR chemistry allowed combinatorial synthesis, where all building blocks could be varied for each compound simultaneously, from our previous experience with molecular glues[21] we were aware

that small changes in even one substituent could have a significant effect on cooperative binding, and this could be difficult to predict. Thus, we decided to vary one building block at a time and keep the rest of the ligand constant to elucidate the SAR step-by-step.

Once compounds were selected for synthesis, synthetic routes were established using commercially available aldehydes and 2-aminopyridines, whereas isocyanides were synthesized from primary aromatic amines (Fig. 1c, d). The selection of building blocks was based on SeeSAR docking, aiming to achieve specific ligand-protein interactions, as explained in detail later. Two general, three-step synthetic routes were developed based on the type of aldehyde building blocks used. In the first synthetic route (route A), Boc-protected aldehydes reacted with 2-aminopyridines and isocyanides in methanol, using scandium triflate as a Lewis acid catalyst to form the GBB intermediate. The Boc-protecting group was removed under acidic conditions, and the chloroacetamide warhead was introduced with an amidation reaction, either with chloroacetyl chloride or an amide coupling. In the second synthetic route (route B), to reduce the cost of certain Boc-protected aldehydes, nitro-substituted aromatic aldehydes were used instead. The GBB-3CR was performed under the same experimental conditions, followed by a nitro-reduction. The nitro-group was reduced either using iron trichloride and zinc or using ammonio-trihydroborate and gold catalysis with Au/TiO$_2$[40]. The two reduction methods led to comparable, almost quantitative yields. The last step, as previously, was the introduction of the electrophilic warhead. Thus, the GBB-3CR allowed multiple combinations of the three main building blocks, facilitating the synthesis of analogs. Synthetic details are provided in the Supplementary Information.

Since ERα is phosphorylated on the penultimate residue, the binding interface to 14-3-3, creates a large, solvent-exposed pocket that can accommodate a small molecule. In general, 14-3-3 molecular glues are expected to stabilize 14-3-3/client complexes based on molecular recognition and shape complementarity. For the 14-3-3/ERα complex, the only accessible contact with ERα is the C-terminal V595, which is located close to the small pocket formed around K122 of 14-3-3, as shown for 127 (Fig. 1a). Thus, for the MCR scaffold, our first modifications focused on the K122 pocket/V595 composite interface. To test whether this scaffold was suitable for the development of 14-3-3σ/ERα stabilizers, we synthesized a small set of derivatives varying only the isocyanide position, which, according to our design, was expected to interact with K122 of 14-3-3. We included two types of isocyanides: benzyl and phenyl isocyanides with a diverse substitution pattern on the aryl ring. The size of the pocket allowed the presence of aromatic rings; since the surrounding 14-3-3 amino acids were primarily hydrophobic, we chose appropriate hydrophobic substituents for SAR exploration. We maintained the covalent chloroacetamide warhead, based on our extensive investigation of electrophiles in our previous work[21].

The compounds were tested in an intact mass spectrometry (MS) assay, which monitored the formation of the covalent bond with C38 of 14-3-3σ. Binding measurements were made in the presence or absence of the phospho-ERα peptide and as a function of time to distinguish between cooperative stabilizers and neutral binders and to select compounds with fast binding kinetics. The phospho-ERα peptide was used at a concentration 2-fold above the dissociation constant (Kd). The assay was performed as a compound titration (Supplementary Figs. 2, 3). We additionally quantified compound binding at 1 µM (10:1 compound: protein) over several time points (Supplementary Table 2). Throughout this work, we reference the time-course data as bar graphs, using grey bars to represent binding to 14-3-3σ alone ("apo") and colored bars to represent binding to 14-3-3 σ/ERα complex.

Testing the first analogs in the MS assay revealed striking differences among benzyl and phenyl analogs (Fig. 1e, Supplementary Fig. 4, Supplementary Table 2). The benzyl analog 1, which lacked aryl ring substitutions, was a neutral binder (for example, in the 8 h measurement, at 1 µM compound we observed 35.7% binding for apo 14-3-3 and 55.7% binding to 14-3-3 in the presence of ERα). Introduction of electron-withdrawing groups, such as halogens or nitro-groups in the o-, m-, or p-position (compounds 2-9) reduced binding in the presence of ERα, even in the case of a small F-substitution. The introduction of halogens and their effect in the different positions was based on previous knowledge; during the optimization of our first series of molecular glues targeting the 14-3-3σ/ERα, the p-Cl substituent was beneficial for cooperative binding[21]. The fact that the SAR did not translate directly between the two series implied potential differences in the binding mode. Interestingly, the non-substituted phenyl analog 10 showed increased binding to 14-3-3σ, but still lacked cooperativity, since it showed increased binding in the presence or absence of ERα (in the 8 h measurement, at 1 µM compound, we observed 56.3% apo binding and 68.9% binding in the presence of ERα). Introduction of halogens in the p-position significantly reduced apo binding, improving compound cooperativity (compounds 11 and 12). For example, for 11 in the 8 h measurement, at 1 µM, compound binding to 14-3-3 alone was 3.4%, and binding in the presence of ERα rose to 24.3%. However, in contrast to the SAR observed in our previous series[21], less binding was observed for the p-Cl analog (11), compared to the non-substituted analog (10). Remarkably, a Me-group in the o-position (13) led to the first molecular glue of the series; compound 13 showed binding in the presence of ERα and very low apo binding (in the 8 h measurement, at 1 µM, compound apo binding was 1.4% and binding in the presence of ERα 42.1%). Bigger substitutions in the o-position significantly decreased the observed binding, indicating unfavorable steric effects (14-16) (Supplementary Fig. 4).

We solved co-crystal structures of the 14-3-3σ/ERα complex with compounds 1, 2, and 10 to elucidate their binding modes (Fig. 1f–i). The crystal structure of compound 1 revealed a comparable binding mode to the original ligand 127, supporting the design and docking hypothesis (Fig. 1f–g). The chloroacetamide moiety of 1 bound covalently to C38 of 14-3-3σ, and two water-mediated hydrogen bonds were formed between the carbonyl group and R41 and the backbone of E115, and one direct hydrogen bond with N42 of 14-3-3 at 3 Å. The imidazopyridine ring was positioned towards the hydrophobic residues (L218, I219, G171, and L222) at the "roof" of the 14-3-3 pocket. In addition, the nitrogen of the five-membered ring formed a hydrogen bond with D215 of 14-3-3, which adopted two conformations upon binding of compound 1. Importantly, a large water network was formed between the benzylamine of 1 to N42, S45 of 14-3-3, which reached the terminal carboxylic acid of V595 and the pT594 of ERα, and K122 of 14-3-3. The benzyl ring, adjacent to the electrophilic warhead was positioned between the hydrophobic 14-3-3 residues F119 and I168, (Fig. 1g). The structural overlay of analogs 1 and 2 showed an identical binding mode with the additional o-F substituent forming a halogen bond with I219 of 14-3-3 (Fig. 1h). Replacing the benzyl ring (1) with a phenyl ring (10) resulted in a slight turn of the compound that placed the aryl rings of both analogs in the same position in the pocket surrounding K122. The water network that connected the amine of compound 1 to residues of 14-3-3 and ERα was conserved upon modification of the benzyl ring (1) to a phenyl ring (10). The removal of the methylene group positioned the aniline nitrogen of (10) closer to the terminal carboxylic acid of ERα, in the position previously occupied by the methylene group of (1). Additionally, it led to increased axial rotation of the bond between the nitrogen and the aryl ring, which might have resulted in the increased apo binding in the MS assay, since this axial rotation was not possible for the benzyl analogs (Fig. 1i).

Since the o-Me substitution correlated with improved binding in the presence of ERα in the MS assay and taking into account the rotational nature of the aniline–phenyl ring bond, we designed and synthesized a series of analogs with double o-substitutions (Fig. 2a, Supplementary Fig. 5). The symmetrical 2,6-di-Me analog 17 showed significantly faster binding in the presence of ERα and almost no apo

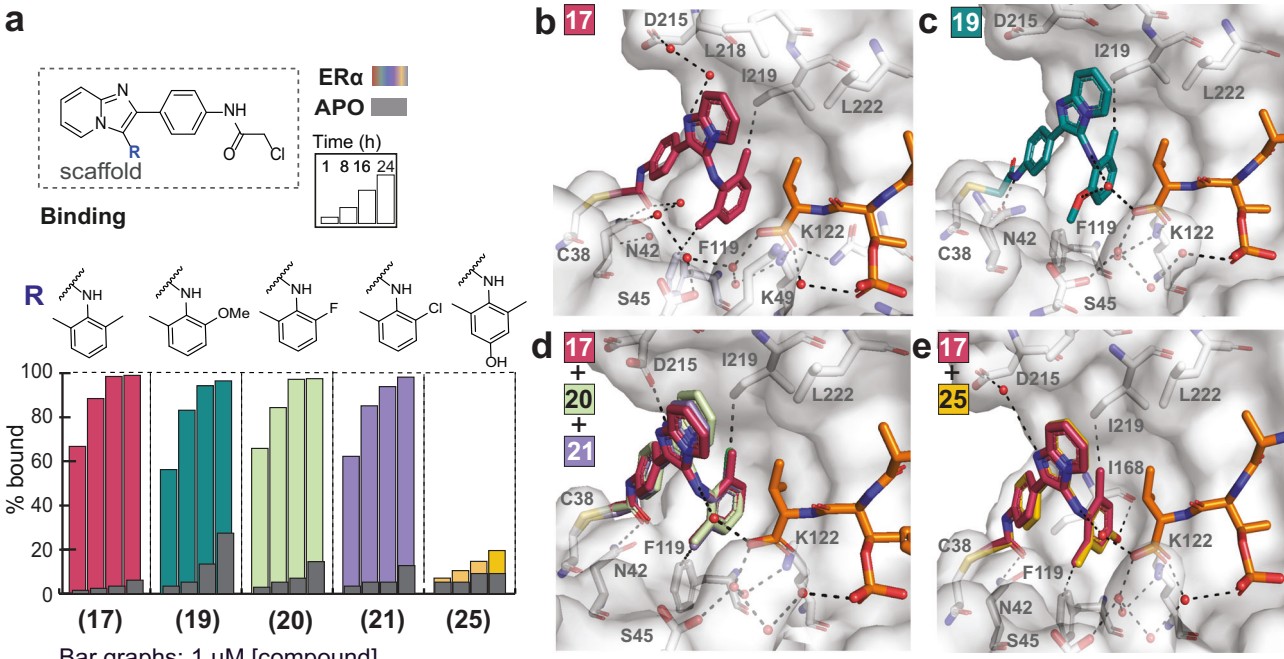

**Fig. 2 | SAR and crystal structures of selected double-*ortho* substituted analogs.** **a** MS bar graphs at 1 μM. For each compound, time course experiments were performed with measurements at 1 h, 8 h, 16 h, and 24 h. ERα data are shown with different colors for each compound, and apo data in grey. **b–e** Crystal structures of 14-3-3σ/ERα with compounds **17** (dark pink sticks) (PDB 9I70), **19** (teal sticks) (PDB 9I71) and overlays of crystal structures for compounds **17** (dark pink sticks) (PDB 9I70), **20** (pale green sticks) (PDB 9I73), **21** (pale purple sticks) (PDB 9I74) and **25** (bright yellow sticks) (PDB 9I75).

binding, indicating molecular-glue-like binding. Analog **18** with an *o*-Et group in addition to *o*-Me, was weaker, indicating unfavorable steric effects (Supplementary Fig. 5). Analogs **19, 20**, and **21** with additional *o*-OMe, *o*-F, and *o*-Cl groups, respectively, showed similar, rapid binding to the symmetric analog **17** and low apo binding. In agreement with our previous observation for the effect of *p*-substitutions (compounds **11** (*p*-Cl) and **12** (*p*-Br)), the introduction of additional substitutions in the *p*-position (-Cl, -Me, -OH) significantly reduced binding (compounds **22-25**), especially for the triple-substituted analogs (double *o*- and *p*-positions).

Co-crystal structures of the 14-3-3σ/ERα complex with compounds **17**, **19**, **20**, **21**, and **25** revealed differences in the positions of the double-*ortho* substituents (Fig. 2b–e). In the case of the symmetric analog **17** one *o*-Me group was oriented in the back of the pocket, which was primarily hydrophobic, forming hydrophobic interactions with I219 (3.2 Å) and facing Val595 of ERα. The second *o*-Me group was oriented in the front of the pocket and formed hydrophobic interactions with F119 (3.8 Å). The formation of these interactions seemed to favorably restrict the axial rotation of the aniline-phenyl bond, locking its position in a highly complementary shape with the composite 14-3-3σ/ERα surface. In analog **19** the larger *o*-OMe group was positioned in the front of the pocket and formed, together with the aniline nitrogen, a water-mediated hydrogen bond with the terminal carboxylic acid of Val595 of ERα. The orientation of the warhead amide differed in two analogs, but in both cases, a direct hydrogen bond with N42 of 14-3-3 was formed (3.0 Å). For the halogen-containing analogs *o*-F (**20**) and *o*-Cl (**21**) the halogens interacted with I219 and the common *o*-Me group with F119, whereas the aniline nitrogen interacted with the terminal carboxylic acid of Val595 of ERα via a water-mediated hydrogen bond. Our NMR data for these compounds do not imply restricted axial rotation. Our assumption is that most of these analogs could be classified as class-1 atropoisomers[41], which are commonly treated as achiral. From a chemical perspective, the rotational bond in these cases can sample the full 360 degrees of rotational conformations about the axis. From a biological perspective, they typically bind

to a given target in only a subset of these conformations, as observed for analogs **17**, **19**, **20**, and **21**, with analogs **19** and **21** adapting the two extreme rotational positions to form favorable interactions with 14-3-3, as demonstrated by crystallography. The binding mode of the triple-substituted analog **25**, although overall comparable with the double-substituted analog **17**, showed an additional hydrogen bond between the *p*-OH group and the backbone carbonyl group of I168 (3.2 Å). The presence of this direct interaction seemed to negatively affect the axial rotation of the aniline-phenyl bond, resulting in significantly reduced binding in the MS assay and loss of cooperativity.

Notably, the alignment of the co-crystal structures of compounds **17** and **25** (Fig. 2e) was almost identical, including the orientation of the common electrophilic warhead. The different interactions based on the compounds' substitution pattern on the *p*-position dramatically affected the observed binding in the mass spec assay (Fig. 2a). Even though the warhead orientation was the same, compound **17** was a fast, cooperative binder in the MS assay, whereas **25** was barely bound. We therefore concluded that the speed of binding was due to cooperativity and shape complementarity, rather than being solely driven by the electrophilic warhead.

We synthesized analogs of compound **21**, maintaining the 2-Cl,6-Me substitutions and varied positions X and Y on the scaffold (Fig. 3a, Supplementary Fig. 6). Position X referred to substitutions on the imidazopyridine ring and was expected to contribute to additional hydrophobic interactions with Val595 of ERα. Position Y included modifications that were expected to improve interactions with the rim of the 14-3-3 pocket, primarily with variations of the aldehyde building blocks and to a smaller extent with different electrophilic warheads. Although modifications in position Y were quite far from the ERα terminal valine, they could result in altered ligand conformations and additional interactions with 14-3-3. In both cases, even the introduction of small groups led to surprising effects in the binding modes, as revealed by crystal structures.

For position X, aiming for hydrophobic interactions with Val595 of ERα, we introduced groups with a range of radii (-F (**26**), -Cl (**27**), -Me

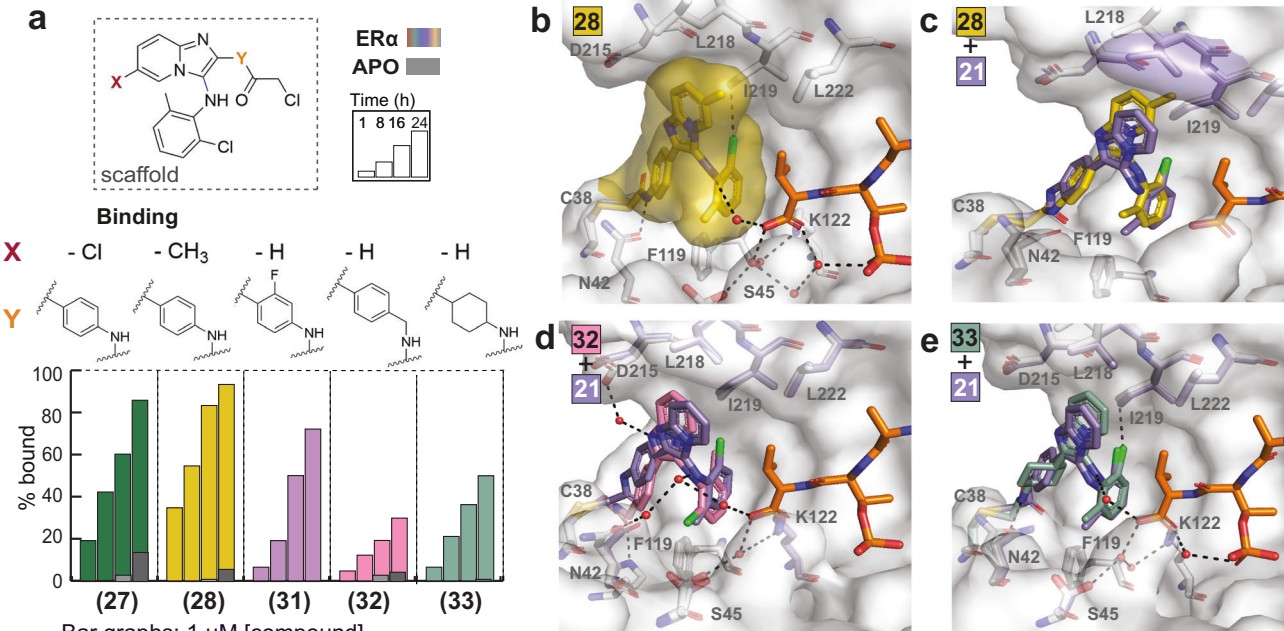

**Fig. 3 | SAR and crystal structures of analogs substituted in positions X and Y.** **a** MS bar graphs at 1 µM. For each compound, time course experiments were performed with measurements at 1 h, 8 h, 16 h and 24 h. ERα data are shown with different colors for each compound, and apo data in grey. **b**–**e** Crystal structure of 14-3-3σ/ERα with compound **28** (dark yellow sticks) (PDB 9I6S), and overlays of crystal structures for compounds **28** (dark yellow sticks) (PDB 9I6S), **32** (pink sticks) (PDB 9I6T) and **33** (emerald green sticks) (PDB 9I6U) with **21** (pale purple sticks) (PDB 9I74).

(**28**), -*i*-Pr (**29**), -CF$_3$ (**30**)). Steric effects seemed to play a bigger role than electronic effects, since the best tolerated substitutions were -Cl (**27**) and -Me (**28**). For position Y, derived from the aldehyde building blocks, the *o*-F analog (**31**) was tolerated, but was weaker compared to the unsubstituted (**21**); a plausible explanation being the rotational nature of the aryl ring–imidazopyridine ring bond, which lost axial symmetry with the presence of the *o*-F group. Introduction of a methylene linker (**32**) or replacement of the aryl ring with a cyclohexyl ring (**33**) that had a similar size made the compounds almost unable to bind in the MS assay. Modifications in the position of the electrophilic warhead were also not tolerated. The less-reactive ester analog (**34**) was inactive, whereas halogenated chloroacetamides (**35-37**) did not lead to the expected mass adducts, showing instability in the MS and unclear chemical reactivity with 14-3-3. A plausible explanation was their increased reactivity, which correlated with reduced stability. The sulfamate warhead (**38**) was tolerated in the MS assay, but was significantly weaker than the chloroacetamide analog, in addition to being less atom efficient.

Crystal structures were solved for **28**, **32**, and **33** and were compared as overlays with analog **21** (Fig. 3b–e). The -Me group on the imidazopyridine ring of **28** was expected to contribute to hydrophobic interactions with Val595 of ERα. However, it disrupted the previously observed binding mode and instead moved the imidazopyridine ring upwards, in the hydrophobic pocket of 14-3-3 formed by L218, I219, and L222. This upward movement affected the conformation of L218, which moved upwards but maintained contact with **28**. The movement in the pocket also altered the interactions of the double-*ortho*-substituted aryl ring; while the *o*-Cl still interacted with I219 on the roof of the binding site, the *o*-Me group moved further away from F119 in the bottom of the pocket, and the interaction was lost. The direct hydrogen bond between the amide and N42 and the water-mediated bond between the aniline nitrogen and the carboxylic acid of Val595 were maintained. Overall, the changes in binding mode were associated with slower binding, but a similar apparent Kd in the MS assay (Fig. S3). Analog **32**, with an additional methylene group on the electrophilic warhead linker, showed a slightly disrupted binding mode. While the

biggest difference was the position of the aryl ring next to the longer linker, the imidazopyridine ring also moved further away from ERα; the latter correlated negatively with the binding observed in the MS assay. In contrast to the previous analogs, the *o*-Cl group on the aryl ring of **32** pointed out of the 14-3-3 pocket and, due to increased distance, was unable to interact with F119. Thus, the presence of an additional methylene group on the linker was sufficient to make the compound unable to stabilize the complex. Analog **33** bearing a cyclohexyl ring instead of an aromatic ring maintained the interaction between the *o*-Cl group of the aryl ring and Ile219, but not the interaction of the *o*-Me group with F119. The orientation of the amide bond next to the warhead also differed; however, the hydrogen bond with N42 was still formed. The presence of the cyclohexyl ring, which had the possibility of adopting more conformations compared to the more rigid aryl ring, correlated with reduced binding to the 14-3-3σ/ERα complex in the MS assay.

Investigation of modifications in positions X and Y provided valuable input in the groups that were tolerated; however, these single-point substitutions did not significantly improve the stabilization effect in the MS assay. Based on the crystallographic input, we synthesized four analogs combining the favorable substitutions from positions X and Y with the symmetric double *o*-Me analog **17**, hypothesizing that a symmetric rotatable bond would correlate with improved potency (Fig. 4a). This hypothesis was readily confirmed by evaluating the symmetric analogs **39-42** in the MS assay. Analogs **39** and **40** included the -Me group on the imidazopyridine ring (X position) and the -F group on the aryl ring (referred to as W position), respectively. Analogs **41** and **42** both had the -F group in W position and a -Me or -Cl group in X position, respectively. All analogs showed comparable, fast, cooperative binding to 14-3-3σ/ERα in the MS assay, indicating that the symmetry of the 2,6-di-Me-aniline ring was more favorable compared to the asymmetric 2-Cl,6-Me. All four analogs showed low apo binding, with analog **41** showing the lowest.

Crystal structures were solved for the symmetric analogs **40**, **41**, and **42**, highlighting the significance of two rotational bonds in the scaffold: the aniline-aryl ring bond, as previously discussed, and the

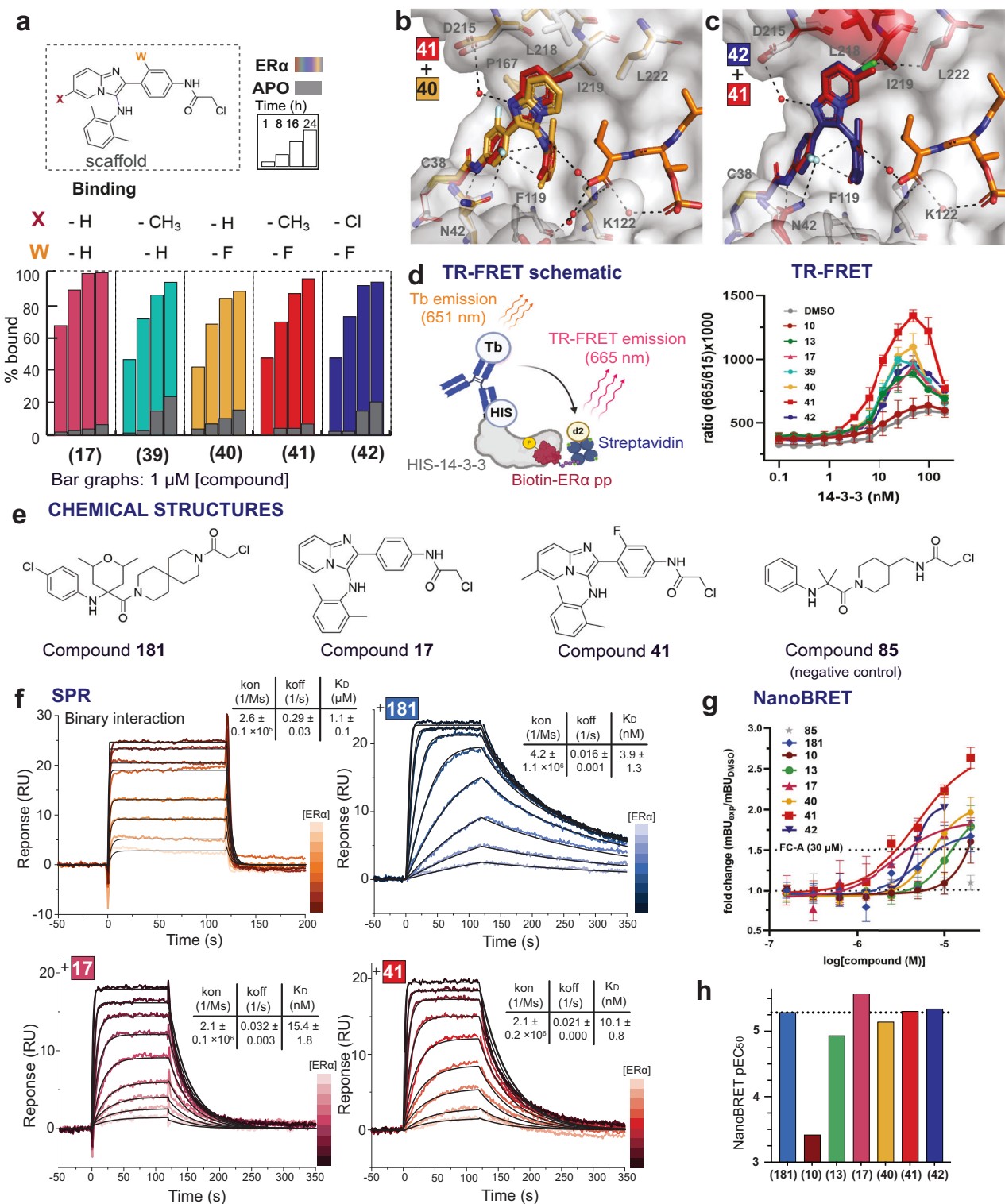

aryl ring–imidazopyridine ring bond (Fig. 4b, c). The o-F substituent in W position, adjacent to the aryl ring-imidazopyridine rotational bond, adopted two different orientations; an inward orientation for analog **40** and an outward orientation for **41** and **42**, which each had an additional substituent on the imidazopyridine ring. With the inward conformation of analog **40**, the o-F substituent interacted with P167 of 14-3-3, whereas for **41** and **42** the o-F group was oriented to make multiple stabilizing interactions, being approximately 3 Å from N42 of 14-3-3, and the o-Me substituent and aniline nitrogen of the

compound itself, thus contributing both to ligand-protein interactions and intramolecular interactions, restricting the axial bond rotation. In agreement with previous observations, the presence of a substituent in the X position (**41** and **42**) led to an upward movement of the imidazopyridine ring, orienting the substituent toward the hydrophobic residues I219 and L222. Overall, the presence of the additional substituent on the imidazopyridine ring in the last two analogs, even though in a distant position compared to the o-F-substituted ring, correlated favorably with the restriction of the rotational bond on the

**Fig. 4 | SAR and crystal structures of 2,6-*di*-Me analogs substituted in positions X and W.** Biophysical data (MS, TR-FRET, SPR) and cell data (NanoBRET). **a** MS bar graphs at 1 μM. For each compound, time course experiments were performed with measurements at 1 h, 8 h, 16 h, and 24 h. ERα data are shown with different colors for each compound, and apo data in grey. **b**, **c** Overlays of crystal structures of 14-3-3σ/ERα ternary complexes with compounds **40** (orange sticks) (PDB 9I6V), **41** (red sticks) (PDB 9I6W), and **42** (blue sticks) (PDB 9I6X). **d** Time-resolved fluorescence energy transfer (TR-FRET) schematic and protein titration data for representative compounds at 100 μM compound or DMSO. For each compound, three independent experiments were performed (*n* = 3). (TR-FRET schematic was created in BioRender. Konstantinidou, M. (2025) https://BioRender.com/yp73j3h). **e** Chemical structures of compounds **181**, **17**, and **41** used in SPR and compound **85**, used in NanoBRET as a negative control. **f** Surface Plasmon Resonance (SPR) data for the binary 14-3-3σ/ERα interaction and ternary interactions with **181**, **17**, and **41** (mean +/− SD, *n* = 2). **g**–**h** 14-3-3σ-HaloTag/NanoLuc-ERα NanoLuciferase bioluminescence resonance energy transfer (NanoBRET) assay in HEK293T cells with compound titrations (1:2 dilution, starting at 40 μM). Data points excluded where compound dosage was toxic to the cells. MCR compounds compared to the previously described stabilizer **181** and **85**, an inactive compound as the negative control. The natural product FC-A was used as a control, dosed at 30 μM (black dashed line). Bar graphs quantifying pEC_{50} values. Data points presented are mean values +/− SD (n = 3 technical) and representative of 4 biological replicates.

F-aryl ring. Additionally, since the 14-3-3σ/ERα composite surface was primarily hydrophobic, the optimized analog **41**, with the additional hydrophobic substituents, achieved favorable shape complementarity (Supplementary Fig. 7). For analogs **41** and **42**, the different substituents of the imidazopyridine ring, -Me (**41**) or -Cl (**42**), did not affect the position of the stabilizer in the crystal structures; however, in the MS assay the -Me group showed lower apo binding to 14-3-3, resulting in higher cooperativity of the ternary complex.

## TR-FRET

Up to this point, the SAR was developed using the intact MS assay and supported by crystallography. In our previous work[20–22], we relied on a fluorescence anisotropy assay (FA) to confirm cooperativity. The FA assay typically used (5-carboxylfluorescein) FAM-labeled phospho-peptides to quantify peptide binding to the compound/14-3-3 complex. For the MCR scaffold, however, this assay proved to be unsuitable, since the extensively conjugated ring system was intrinsically fluorescent in the same wavelengths as the FAM-labeled peptide (480 nm and 520 nm). To circumvent this issue, we turned our attention to far-red fluorescent dyes. The cy5-labeled ERα-peptide with excitation wavelength of 651 nm and emission wavelength of 670 nm, significantly affected the Kd of the 14-3-3/ERα complex. The reported Kd using the acetylated ERα peptide in an isothermal calorimetry (ITC) experiment or the FAM-labeled-ERα in an FA assay is in the range of 1-2 μM[21]. The cy5-ERα-peptide, however, resulted in a Kd of 6 nM, indicating significant dye binding (Supplementary Fig. 8).

As a suitable alternative to the FA assay, we developed a time-resolved fluorescence energy transfer (TR-FRET) assay[42], using HIS-tagged-full-length 14-3-3σ, biotin-labeled-ERα peptide, an anti-HIS-tag monoclonal antibody conjugated with a Tb(III) cryptate as the donor, and streptavidin conjugated with the D2 dye as the acceptor. The observed Kd for this system was 30 nM. To ensure that the observed difference in Kd from ITC or the FAM-labeled-ERα in an FA assay was not related to the biotin-tag on the peptide, we performed a competition assay with the FAM-labeled peptide. The Kd of the biotin-peptide was 2.5 μM, in good agreement with the FAM-peptide's Kd (Supplementary Fig. 8). This result suggested that the lower Kd in the TR-FRET assay was due to avidity[43]; since 14-3-3 and the antibody were both dimers and streptavidin was tetravalent, different multivalent complexes could form, resulting in differences in the observed binding affinity between the labeled and the unlabeled proteins.

We performed assay optimization with 2D-titrations of 14-3-3, biotin-ERα, and donor/acceptor ratios. We then tested the synthesized compounds using the optimized experimental conditions: 200 nM 14-3-3 (top concentration, 2-fold dilution), 50 nM biotin-ERα, 0.166 nM MAb anti-6HIS Tb, and 6.25 nM SA-D2. The compounds were tested at 100 μM, and DMSO was used as a negative control. 14-3-3 was titrated using an Echo acoustic dispenser, followed by the addition of the compounds and the peptide; after 1 h incubation at room temperature, the donor and acceptor were added. In contrast to the MS assay, maximum signal was obtained after 2 h rather than 16 h–a plausible

explanation again being avidity. The tight Kd in the TR-FRET assay resulted in a small assay window, and a hook effect was observed at higher protein concentrations (ca. 50 – 100 nM 14-3-3).

Nevertheless, the TR-FRET assay was sensitive to the addition of molecular glues. In addition to quantifying fold stabilization for the 14-3-3σ/ERα complex (*AppKd(DMSO)*/*AppKd(compound)*), we also quantified the fold increase in the TR-FRET signal (the ratio of observed Emax and Emin). To validate the TR-FRET assay, we included two positive controls: the natural product FC-A and our previously described stabilizer compound **181**[21]. The two compounds gave comparable results. FC-A had an *AppKd(compound)* of 11 nM, fold-stabilization of 3.09, and fold-increase of 5.52. Compound **181** had an *AppKd(compound)* of 8.6 nM, fold-stabilization of 3.95, and fold-increase of 5.52 (Supplementary Fig. 9, Supplementary Table 3). In good agreement with the MS data, neutral binders **1** and **10** showed only a small signal shift compared to the DMSO control (Fig. 4d, Supplementary Fig. 9). The 2-Me analog **13** showed 4.19-fold-stabilization and 2.36-fold increase, whereas 2,6-di-Me analog **17** showed 4.78-fold-stabilization and 2.58-fold increase, the same rank order as the MS assay. The more sterically hindered analog **18** (*o*-Me, *o*-Et) showed weaker stabilization (3.06-fold stabilization and 2.14-fold increase). Analogs **19**, **20**, and **21**, all bearing the *o*-Me group and additional *o*-OMe, *o*-F, and *o*-Cl groups, respectively, showed comparable stabilization (4.41 – 5.23-fold stabilization and 2.1 – 2.48-fold increase). The highest stabilization effect was observed for the symmetric analogs **39** – **42**. Analog **39** with a -Me group in X position had an *AppKd(compound)* of 7.8 nM, fold-stabilization of 4.35, and fold-increase of 2.61, whereas **40** with the -F group in W position had an *AppKd(compound)* of 8.4 nM, fold-stabilization of 4.04, and fold-increase of 3.12. Analog **41**, which had both the -Me group in X position and the -F group in the W position, had the lowest *AppKd(compound)* (5.2 nM) and the highest fold-stabilization (6.53) and fold-increase (3.71) of the series. Analog **42**, which differed only in the X position (-Cl instead of -Me group), appeared weaker (*AppKd(compound)* 8.8 nM, fold-stabilization of 3.87 and fold-increase of 2.97). In summary, while the fold-changes were dampened by avidity, 14-3-3/ERα molecular glues showed the same rank-order in the TR-FRET assay as in the mass spectrometry assay used for initial SAR.

## SPR

Surface Plasmon Resonance (SPR) was then used to analyze the kinetic parameters of the ERα peptide binding to 14-3-3σ in the presence of compounds **181**, **17**, and **41**, and to compare the *AppKd(compound)* and kinetics to the binary ERα/14-3-3σ interaction (Fig. 4e, f, Supplementary Figs. 10, 11, Supplementary Table 4). Here, 14-3-3σ fused with a Twinstrep-tag was captured on a SPR chip coated with Strep-Tactin XT, after which a 2-fold dilution series of acetylated ERα phospho-peptide was injected. For the binary interaction, the fast dissociation rate (k_{off}) reached the limit of detection of the SPR instrument, due to a relatively weak interaction with a Kd of 1.1 μM, which was in line with the ITC and FA experiments. The covalent bond between the chloroacetamide warhead of the compounds and C38 of 14-3-3σ was formed after overnight incubation in the presence of ERα. Immobilization of this complex on the chip, followed by extensive washing to remove the

bound ERα peptide, allowed us to determine the kinetics of ERα binding to the 14-3-3σ/stabilizer complex. The previously described stabilizer **181** decreased the off-rate to $0.016\,s^{-1}$ and simultaneously increased the association rate ($k_{on}$) by a factor of 16, resulting in a low nanomolar affinity constant (3.9 nM; stabilization = 282-fold). The analogs of the newly designed scaffold, **17** and **41**, both showed an 8-fold increase in association rate compared to the binary interaction. The binding of compound **41** induced a stronger decrease in the dissociation rate of ERα compared to **17**, which resulted in Kd values of 10.1 nM and 15.1 nM for **41** and **17**, respectively. The higher stabilizing potency of **41** compared to **17** (stabilization = 110-fold for **41** and 71-fold for **17**) were in rank-order agreement with the TR-FRET data. The decreased dissociation rates in the presence of the stabilizers, especially for **41** and **181**, increased the residence time of ERα binding to 14-3-3 from approximately 3.4 s to 47.6 s and 62.5 s, respectively.

## NanoBRET

To test the effects of the compounds on the full-length PPI, we used a NanoLuciferase bioluminescence resonance energy transfer (Nano-BRET) assay we developed previously[44]. Compounds were tested using a C-terminal fusion 14-3-3σ-HaloTag and full length, N-terminal fusion NanoLuc-ERα (Fig. 4g, h; Supplementary Table 5). Briefly, NanoLuc-ERα and 14-3-3σ-HaloTag plasmids were transfected in 1:10 ratio in hormone-deprived HEK293T cells. After 48 hours post-transfection, cells were seeded in assay plates with the experimental wells treated with 100 nM HaloTag NanoBRET 618 ligand (Promega) and no-ligand control wells treated with DMSO (v/v). Cells were treated for 24 hours with compounds in 1:2 dilution series starting at 40 μM. The BRET signal was read and normalized against DMSO-treated samples. All active compounds resulted in an increase in BRET signal compared to the negative control, **85**. The previously described compound **181**[21] stabilized the 14-3-3σ/ERα complex in cells with an $EC_{50}$ value of 5.2 μM and a 1.7-fold increase in the BRET signal, which was higher than the observed BRET signal with the natural product FC-A, tested at 30 μM. Of the MCR compounds tested, the neutral binder **10** was less effective than **181** ($EC_{50}$ value > 100 μM, 1.6-fold increase). Compound **13** showed an improved $EC_{50}$ and fold-increase compared to **10** (12 μM and 1.8-fold). Compound **17** had the lowest $EC_{50}$ value of 2.7 μM and showed a 1.8-fold increase in BRET signal, a slight improvement compared to **181**. Compounds **40** and **42** had the same 2-fold-increases in BRET signal, though they exhibited slightly different $EC_{50}$ values (7.2 and 4.6 μM, respectively). Compound **41** resulted in the highest increase in BRET signal, a 2.6-fold increase; however, it had a similar $EC_{50}$ value to **181** of 5 μM. In the NanoBRET assay, compounds **17** and **41** performed most effectively, based on the $EC_{50}$ value (**17**) and fold-increase in BRET signal (**41**) of the panel of compounds tested. The majority of MCR stabilizers showed similar $EC_{50}$ values to the previous scaffold represented by **181**, but consistently showed improved fold-increase in BRET signal.

Compounds **181**, **17**, and **41**, along with the negative control, **85**, were tested in a NanoBRET assay where the cysteine of interest in 14-3-3σ C38, was mutated to an asparagine, 14-3-3σ$^{C38N}$-HaloTag (Supplementary Fig. 12a). The BRET signal did not increase for **85**, **181**, or **17** with increasing compound concentration. There was a minimal increase in BRET signal for compound **41**, from 2.5 to 3.5 mBU at 20 μM **41**. In comparison to the assays done with 14-3-3σ$^{WT}$-HaloTag, the non-normalized BRET signal increased from 6.7 to 16.1 mBU at the same concentration of **41** (data shown as normalized). The natural product FC-A was also tested as a positive control; however, since the compound is non-covalent, the effect was not specific to the 14-3-3 sigma isoform, and an increase in BRET signal was observed for both constructs (Supplementary Fig. 12b). In summary, when C38 was not present in 14-3-3σ, the covalent compounds were unable to bind and stabilize the full-length 14-3-3σ/ERα complex in cells (Supplementary Fig. 12).

## Discussion

Using MCR chemistry, we demonstrated the potential of a scaffold-hopping approach for molecular glues stabilizing a native 14-3-3/client PPI. Our approach combined computational de novo design with multi-component reaction chemistry, which included short, efficient synthetic routes with multiple points of variation, accelerating the synthesis of analogs. Thus, two distinct optimization approaches led to two highly diverse chemical series, converging on similar efficacies as 14-3-3σ/ERα molecular glues. It is noteworthy that the natural product FC-A stabilizes the 14-3-3/ERα complex to a similar extent[21]. Since ERα terminates at the FC-A pocket, only the C-terminal valine residue is available for binding; it may be that these three distinct series have maximized the binding energy available for gluing this protein-protein interaction.

In the MCR approach, structure-activity relationships were established with an intact mass spectrometry assay, which allowed the distinction between neutral binders and stabilizers – cooperative molecular glues. Overall, the SAR showed that the introduction of even small substituents in the case of molecular glues could profoundly affect their potency and cooperativity. Crystal structures of ternary complexes were crucial in elucidating small changes in the binding modes of otherwise highly similar analogs. Starting from the weak neutral binder **1** and by removing one methylene group, we significantly increased binding to the complex for compound **10**. Introduction of substituents in the o-position was sufficient to reduce apo binding and turn the compounds from neutral binders to cooperative molecular glues. Although the number of rotational bonds is generally kept to a minimum, in this case, we took advantage of two rotational bonds in the scaffold and with appropriate substituents achieved favorable ligand conformations, resulting in increased potency, as shown for analogs **17** and **41**. Notably, most of the synthesized analogs contained the same electrophilic warhead but varied significantly in their binding and stabilization potential. Thus, we were able to demonstrate that binding was not solely driven by the covalency, but rather that specific, favorable interactions of the ligands in the PPI composite surface contributed to molecular recognition and shape complementarity. Additionally, since the 14-3-3σ/ERα interface was largely hydrophobic, most of the favorable ligand modifications were non-polar, indicating that the overall shape complementarity was a key factor in improving stabilization of the complex.

Several biophysical assays provided complementary strategies for evaluating these novel molecular glues. A TR-FRET assay circumvented the issue of scaffold fluorescence, which could have been a limiting factor in establishing SAR and allowed the rank-ordering of analogs. A new SPR assay, in which the covalent ligand was pre-associated with 14-3-3σ before immobilization, provided an insightful analysis of binding/unbinding kinetics and started to hint at differences between the two series of molecular glues. Taken together, biophysical assays allowed the quantification of cooperativity (MS, TR-FRET) and kinetics (SPR); importantly, the observed SAR was consistent among these three assays. The NanoBRET assay further showed good correlation with the biophysical protein/phosphopeptide assays while demonstrating stabilizing of the full-length proteins in intact cells.

Overall, we showed that the compounds that were binding the fastest and at the lowest concentrations in the presence of the peptide in the mass spectrometry assay also showed the largest stabilization by TR-FRET and NanoBRET. Stabilization was further validated and explained by 100 + -fold reductions in Kd for the protein-peptide interaction, driven mostly by reduced off-rates. Thus, the observed ranking and SAR of the compounds translated across different assay formats. Thorough characterization and validation of molecular glue scaffolds in orthogonal assays were beneficial for rational scaffold optimization. Structural studies indicated minor changes in the shapes of the molecules but large changes in stabilization that correlated closely with improved interactions with the protein and shape complementarity. Taken together, the optimized, cell-active MCR scaffold

will facilitate chemical biology approaches to study the 14-3-3/ERα interaction, which has so far been unexploited for drug discovery. We also propose that the scaffold-hopping strategy will be broadly applicable in the rapidly evolving field of molecular glues, where we are just beginning to explore 'glue-like' chemical space.

## Methods

### Protein expression and purification

The 14-3-3σ isoform (full-length for mass spectrometry and TR-FRET assays, ΔC for crystallography) with an N-terminal His6 tag was expressed in Rosetta™ 2(DE3)pLysS competent *E. coli* (Novagen) from a pPROEX HTb expression vector. After transformation, following manufacturer's instructions, single colonies were picked to inoculate 30 mL precultures (LB), which were added to 1.5 L terrific broth (TB) medium after overnight growth at 37 °C, 250 rpm. Expression was induced upon reaching $OD_{600}$ 1.9–2.1 by adding 400 μM IPTG. After overnight expression at 30 °C, 150 rpm, cells were harvested by centrifugation at 6500 rpm, resuspended in lysis buffer (50 mM HEPES pH 7.5, 500 mM NaCl, 20 mM imidazole, 10% glycerol, 1 mM TCEP), and lysed by sonication. The His6-tagged protein was purified by Ni-affinity chromatography (Ni-NTA Agarose, Invitrogen) (Wash buffer 50 mM HEPES pH 7.5, 500 mM NaCl, 20 mM imidazole, 1 mM TCEP; Elution buffer 50 mM HEPES pH 7.5, 500 mM NaCl, 500 mM imidazole, 1 mM TCEP) and analyzed for purity by SDS-PAGE and Q-Tof LC/MS. The protein was buffer exchanged (Storage buffer 25 mM HEPES pH 7.5, 150 mM NaCl, 1 mM TCEP) and concentrated to ~16 mg/mL and aliquots flash-frozen for storage at −80 °C. The ΔC variant was truncated at the C-terminus after T231 to enhance crystallization and after the first Ni-affinity chromatography column, the construct was treated with TEV protease to cleave off the His6 tag during dialysis (25 mM HEPES, pH 7.5, 200 mM NaCl, 5% glycerol, 10 mM $MgCl_2$, 250 μM TCEP) overnight at 4 °C. The flow-through of a second Ni-affinity column was subjected to a final purification step by size exclusion chromatography (Superdex 75 pg 16/60 size exclusion column (GE Life Science) (SEC buffer: 25 mM HEPES pH 7.5, 100 mM NaCl, 10 mM $MgCl_2$, 250 μM TCEP). The protein was concentrated to ~60 mg/mL, analyzed for purity by SDS-PAGE and Q-Tof LC/MS and aliquots flash-frozen for storage at -80 °C.

The 14-3-3 σ isoform (FL) with an N-terminal His6-tag and C-terminal Twinstrep-tag was expressed in BL21(DE3) competent *E. coli* (Novagen) from a pET28a expression vector. After transformation, following manufacturer's instructions, single colonies were picked to inoculate 30 mL precultures (LB), which were added to 1.5 L TB medium after overnight growth at 37 °C, 250 rpm. Protein expression was induced at $OD_{600}$ ~ 0.8 by adding isopropyl β-D-1-thiogalactopyranoside (IPTG; 0.4 mM) and cells were harvested by centrifugation (10 min, 4 °C, 16,000 × g) after overnight expression (18 °C, 140 rpm). Pellets were resuspended in wash buffer (50 mM Tris pH 8.0, 300 mM NaCl, 12.5 mM imidazole and 2 mM β-mercaptoethanol (βME)). After homogenizing the cells (40 bar, Emulsiflex-C3 homogenizer), the soluble fraction was collected by centrifugation (30 min, 4 °C, 40,000 × g) and loaded onto a Ni2+-affinity column pre-equilibrated with wash buffer. After a washing step (wash buffer + 20 mM imidazole), the bound protein was eluted with 200 mM imidazole. After Ni-affinity chromatography the elution fraction was loaded on a StrepTactin XT column (Iba Lifesciences) (wash buffer 100 mM Tris pH 8, 150 mM NaCl, 1 mM EDTA; elution buffer 100 mM Tris pH 8, 150 mM NaCl, 1 mM EDTA, 50 mM biotin) and analyzed for purity by SDS-PAGE and Q-Tof LC/MS. The protein was buffer exchanged (storage buffer 25 mM HEPES pH 8, 100 mM NaCl, 10 mM $MgCl_2$, 0.5 mM TCEP) and concentrated to 34.8 mg/mL and aliquots flash-frozen for storage at −80 °C.

### Peptides

Peptides for mass spectrometry, fluorescence anisotropy, and TR-FRET assays were purchased from Elim Biopharmaceuticals, Inc.

(Hayward, CA). Peptides for SPR and X-ray crystallography were purchased from GenScript Biotech Corp. The following peptides were used:

*Ac*-KYYITGEAEGFPA{pT}V-*COOH* (MS assay, 15-mer),
*5-FAM*-AEGFPA{pT}V-*COOH* (FA assay, 8mer ERα-pp),
*cy5*-KYYITGEAEGFPA{pT}V-*COOH* (FA assay, 15-mer),
*biotin*-KYYITGEAEGFPA{pT}V-*COOH* (TR-FRET assay, 15-mer),
*Ac*-EGFPA{pT}V-*COOH* (crystallography and SPR, 7-mer)

### Intact mass spectrometry assay

Mass spectrometry dose response assays were performed on a Waters Acquity UPLC/ Xevo G2-XS Q-Tof mass spectrometer. A Waters UPLC Protein BEH-C4 Column (300 Å, 1.7 μm, 2.1 mm×50 mm) was used to desalt the samples prior to application on the mass spectrometer. For 19-point MS dose responses, 50 mM compound stocks in DMSO were serially diluted in 3-fold increments in a master plate, then 1000 nL of the compounds were transferred to the assay plates. Master mixes containing 100 nM full-length wild-type 14-3-3σ in the absence or presence of 2 μM ERα were then dispensed into 384-well plates (Greiner Bio-One, catalog number 784201). Assay buffer was TRIS (10 mM, pH 8.0), and final volume per well was 50 μL, with final top concentration of compounds dose response series at 1 mM. The reaction mixtures were incubated for 1 h at rt before subjected to MS. Four measurements (1 h, 8 h, 16 h, 24 h) were performed for time-course experiments. The injection volume for each sample was 6 μL. 24 μL of sample was needed for the time-course experiments, so the total volume in the assay plate was adjusted to 50 μL, to account for the dead volume in the injections. Data collection and automated processing followed a custom workflow, as previously described[45]. z Plots were created using GraphPad Prism with the log(agonist) vs. response (variable slope, four parameters) fitting model.

### $K_d$ Determination for FAM-, cy5- And biotin-labeled ERα peptides

For $K_d$ determination, N-terminal fluorescein-labeled ERα peptide (5-FAM) or cy5-labeled ERα peptide and HIS-tag FL 14-3-3σ were diluted in buffer (10 mM HEPES pH 7.5, 150 mM NaCl, 0.05% tween 20, 0.05% BGG (bovine gamma globulin)). Two-fold dilution series of 14-3-3 were made in black, round-bottom 384-microwell plates (Greiner Bio-one 784900) in a final sample volume of 10 μL in triplicates. FAM- or cy5-labeled ERα peptides (final assay concentration 10 nM) were dissolved in assay buffer and mixed with the protein dilution series on the plates. Fluorescence anisotropy measurements were performed after 1 h incubation at room temperature on an Envision HTS Dual Detector 2105 plate reader (for FAM-labeled ERα peptide: filter set lex: 480, lem: 535, and D505fp/D535 advanced dual mirror). For cy5-labeled ERα peptide filter set lex: 620, lem: 688 nm, and D658fp/D688 advanced dual mirror. Data were reported at the endpoint. Prism 10 (GraphPad) was used to generate plots using the [agonist] vs. response (variable slope, four parameters) fitting model to determine $K_d$ values.

For $K_d$ determination of the biotin-labeled ERα peptide, a competition assay was performed. 5-FAM- and biotin-labeled ERα peptide were diluted in buffer (10 mM HEPES pH 7.5, 150 mM NaCl, 0.05% tween 20, 0.05% BGG (bovine gamma globulin)). Two-fold dilution series of biotin-labeled ERα peptide were made in black, round-bottom 384-microwell plates (Greiner Bio-one 784900) in a final sample volume of 10 μL in triplicates. A mastermix of 14-3-3σ and 5-FAM-ERα was dispensed on the assay plate (final assay concentrations: 6 μM 14-3-3σ ($IC_{80}$) and 10 nM 5-FAM-ERα). Fluorescence anisotropy measurements were performed after 1 h incubation at room temperature using a Molecular Devices ID5 plate reader (filter set lex: 485 ± 20 nm, lem: 535 ± 25 nm; integration time: 50 ms; settle time: 0 ms; shake 5 sec, medium, read height 3.00 mm, G-factor = 1). Data were reported at endpoint. Prism 10 (GraphPad) was used to generate plots using the

[agonist] vs. response (variable slope, four parameters) fitting model to determine $K_d$ values. The obtained $K_d$ value was corrected using the Cheng-Prusoff Eq. (1).

$$K_d = IC_{50}/(1 + [S]/K_m)$$
$$K_d = 6.9/(1 + 50\,nM/30\,nM) = 2.5\,\mu M \tag{1}$$

## TR-FRET protein titrations

For assay optimization, 2D titrations of biotin-labeled ERα peptide, HIS-tag FL 14-3-3σ and streptavidin-D2 were performed in assay buffer (10 mM HEPES pH 7.5, 150 mM NaCl, 0.05% tween 20, 0.05% BGG (bovine gamma globulin)). The donor (MAb anti-6HIS Tb cryptate gold, (Revvity, 5000 tests #61HI2TLA)) concentration was kept constant at 0.166 nM. For TR-FRET protein titrations, biotin-labeled ERα peptide (50 nM), the compounds or DMSO (100 μM), MAb anti-6HIS Tb cryptate gold (0.166 nM) and streptavidin-D2 (6.25 nM) were mixed in assay buffer (10 mM HEPES pH 7.5, 150 mM NaCl, 0.05% tween 20, 0.05% BGG (bovine gamma globulin)). 2-fold serial dilutions of HIS-tag FL 14-3-3σ were performed (200 nM top assay concentration, 12-point dilution series). The assay was performed in 384-well microplates (Corning 4513, low volume white) at a volume of 10 μL per well. The following procedure was used: The compounds (50 mM stocks in DMSO) were transferred to echo LDV masterplates. 20 nL were transferred from the masterplate to the assay plate to achieve 100 μM compound concentration in the assay using Echo acoustic dispensing. The biotin-labeled ERα peptide was dissolved in assay buffer (10 mM HEPES pH 7.5, 150 mM NaCl, 0.05% tween 20, 0.05% BGG (bovine gamma globulin)) and dispensed in the assay plate using Dragonfly. 14-3-3 dilution series were prepared using Echo acoustic dispensing. Assay plates were incubated for 1 h at room temperature before the addition of a mastermix containing the donor (MAb anti-6HIS Tb cryptate gold) and acceptor (streptavidin-D2) in assay buffer. The mastermix was dispensed with Dragonfly. Assay plates were incubated for 2 h at room temperature prior to TR-FRET measurements using the Envision HTS Dual Detector 2105 plate reader equipped with the TR-FRET filter set (320/615/665 nm) and a D407/D630 advanced dual mirror. A 50 μs delay was employed to reduce background fluorescence. The TR-FRET signal was obtained through calculating the ratio of 665 nm to 615 nm fluorescence (x 1000), and Prism 10 (GraphPad) was used to generate plots using the [agonist] vs. response (variable slope, four parameters) fitting model. For each compound three independent experiments were performed ($n = 3$).

## SPR

The SPR experiments were performed at 25 °C using a Biacore X100 (Cytiva) and a 200 nm Strep-Tactin XT derivatized linear polycarboxylate hydrogel chip, medium charge density (XanTec Bioanalytics). All proteins and peptides were dissolved in fresh running buffer prepared with ultrapure water and filtered through a 0.2 μm filter (10 mM HEPES pH 7.4, 200 mM NaCl, 50 μM EDTA, 0.005% P20). First the surface was conditioned with a 1 min injection of 3 M Guanidine HCl. Then, the recombinant 14-3-3σ-Twinstrep protein (250 nM) was captured on flow cell 2 of the sensor chip at a flow rate of 10 μL/min for 2 minutes, which resulted in a capture level of 1000 RU. For the ternary interaction, 14-3-3σ-Twinstrep protein (250 nM) was first incubated overnight with 1 μM Ac-ERα peptide and 20 μM compound prior to immobilization to the chip (similar flow rate and injection time used as for apo 14-3-3). The bound ERα peptide was washed away using running buffer that flowed over the chip for 15 min at flow rate of 30 μL/min. Flow cell 1 was left blank as a reference surface. After immobilization of the protein, the Biacore X100 was primed with running buffer. Multi-cycle kinetic measurements were conducted at a flow rate of 30 μL/min. A 2-fold dilution

series of analyte (Ac-ERα peptide) in running buffer was injected over the sensor chip for 2 min, followed by dissociation for 3 min (binary interaction), 7 or 13 minutes (ternary interaction) until full dissociation of the peptide. For the binary interaction, the highest concentration of ERα was 50 μM, and for the ternary interactions, this was 250 nM. Between cycles of one multi-cycle measurement, no regeneration step was performed due to the complete dissociation of the analyte. After a measurement, the chip was regenerated by 2 times 30 sec injections at flow rate 10 μL/min of 3 M Guanidine HCl. The data collection rate of the instrument is 1 datapoint/sec. The data was corrected by double subtracting to the reference surface (flow cell 1) and buffer injection (blank), and analyzed using 1:1 interaction fitting model with the BIA evaluation software (2020). All parameters were fitted using global fitting, including the Rmax, because a single immobilization was used for all analyte concentrations. The RI was set constant at 0 RU. Next to kinetic fitting, the affinity fit was made using a 1:1 binding model by plotting the response (RU) at steady state over the analyte concentration (SI Fig. 11). This was not possible for the data with compound **181** since steady state was not reached for all analyte concentrations.

## X-ray crystallography data collection and refinement

The 14-3-3σΔC protein, acetylated ERα, and compounds (50 mM stock in DMSO) were dissolved in complexation buffer (25 mM HEPES pH=7.5, 2 mM MgCl₂, and 100 μM TCEP) and mixed in a 1:2:3 or 1:2:5 molecular stoichiometry (protein: peptide: compound) with a final protein concentration of 12 mg/mL. The complex was set up for sitting-drop crystallization after overnight incubation at 4 °C, in a custom crystallization liquor (0.05 M HEPES (pH 7.1, 7.3, 7.5, 7.7), 0.19 M CaCl₂, 24-29% PEG400, and 5% (v/v) glycerol). Crystals grew within 10-14 days at 4 °C. Crystals were fished and flash-cooled in liquid nitrogen. X-ray diffraction (XRD) data were collected at the European Synchrotron Radiation Facility (ESRF Grenoble, France, beamline ID23-1, ID30A-3/MASSIF-3, or ID23-2) or at the Deutsches Elektronen-Synchrotron (DESY Hamburg, Germany, beamline PETRA III). Data was processed using CCP4i2 suite (version 8.0.019). After indexing and integrating the data, scaling was done using AIMLESS. The data was phased with MolRep, using PDB 4JC3 as template. The presence of co-crystallized ligands was verified by visual inspection of the Fo-Fc and 2Fo-Fc electron density maps in COOT (version 0.9.8.93). If the electron density corresponding to the co-crystallized ligand was present, its structure, restraints, and covalent bond were generated using AceDRG. After building in the ligand, model rebuilding and refinement were performed using REFMAC5. The PDB REDO server (pdb-redo.edu) was used to complete the model building and refinement. The images were created using the PyMOL Molecular Graphics System (Schrödinger LLC, version 4.6.0). See Supplementary Table 6 for data collection and refinement statistics.

## NanoBRET

NanoBRET assays were performed as previously described[44]. HEK293T cells were cultured DMEM, high glucose (Gibco) supplemented with 10% charcoal stripped Fetal Bovine Serum (FBS; Gibco) and 1% penicillin/streptomycin. Cells were transfected with a 1:10 ratio of Nanoluc-ERa:14-3-3σ-HaloTag plasmid for 48 hours using jetOPTIMUS transfection reagent (Polyplus). Cells were then seeded at 8000 cells per well in a 384-well plate (Corning #3570) in FluoroBrite DMEM (phenol red-free; Gibco) with 4% charcoal stripped FBS and treated with 100 nM HaloTag NanoBRET 618 Ligand (Promega) or an equivalent volume of DMSO as a no ligand negative control. Following plating, cells were treated for 24 hours with compound in 1:2 dilution series starting at 40 μM (0.35% DMSO final concentration). After 24 hours, the BRET signal was read using an EnVision XCite 2105 plate reader at 618 nm (HaloTag) and 460 nm (NanoLuc). The final corrected NanoBRET ratio was calculated using the

following Eq. (2):

$$Corrected\ BRET\ ratio = \left(\frac{618\,nm}{460\,nm}\right)_{HaloTag\ Ligand}$$
$$- \left(\frac{618\,nm}{460\,nm}\right)_{No\ ligand\ control} \quad (2)$$

The BRET ratios were normalized to samples treated with DMSO.

## Docking
Computational design for SAR optimization and docking was performed with SeeSAR version 14.0.0; BioSolveIT GmbH, Sankt Augustin, Germany, 2022, www.biosolveit.de/SeeSAR.

## Software versions
Illustrator (22.1 (64-bit)), Biorender (64-bit), Pymol (4.6.0), CCP4i2 (8.0.003), COOT (0.9.8.1), Phenix (1.19.2-4158), GraphPad Prism (10.2.1), BIA evaluation software (2020).

## Reporting summary
Further information on research design is available in the Nature Portfolio Reporting Summary linked to this article.

## Data availability
All data generated or analyzed during this study are included in this published article (and its supplementary information files). The mass spec, TR-FRET, SPR, and NanoBRET data generated in this study are provided in the Supplementary Information/Source Data file. Supplementary Figs. and tables, synthetic procedures, compound characterization, NMR spectra, and crystallography data are provided in the supporting information (PDF). The crystal structures were deposited in the protein data bank (PDB) under accession codes: 9I6S (compound 28), 9I6T (compound 32), 9I6U (compound 33), 9I6V (compound 40), 9I6W (compound 41), 9I6X (compound 42), 9I6Y (compound 1), 9I6Z (compound 2), 9I70 (compound 17), 9I71 (compound 19), 9I72 (compound 10), 9I73 (compound 20), 9I74 (compound 21), 9I75 (compound 25), 8ALW (compound 127) Densities, PDB IDs and omit density maps are provided in supplementary Figs. 13, 14. Source data are provided with this paper.

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

## Acknowledgements

This research was funded by the Ono Pharma Foundation Breakthrough Science Initiative Award (M.R.A.) and NIH/NIGMS GM147696 (M.R.A.). Funding was received from the NIH for use of the Bruker SPR 24 Pro (S10 OD030265). (M.R.A.) Funding was received from the Netherlands Organization for Scientific Research (NWO) through the Gravity program 024.001.035 (L.B.) and ENW M-grant OCENW.M20.200 (L.B.) We acknowledge Foundation for Research and Innovation (H.F.R.I.) under the "2nd Call for H.F.R.I. Research Projects to support Post-Doctoral Researchers" (Project Number: 0911) and Empeirikion Idryma (C.G.N) We thank Amanda Paulson for the automated mass spec data processing infrastructure in the SMDC. We acknowledge the European Synchrotron Radiation Facility (ESRF) for the provision of synchrotron radiation facilities, and we would like to thank David Flot and Max Nanao for assistance and support in using beamlines ID23-1, ID23-2, ID30A-3 (mx2407 and mx2526). We thank DESY (Hamburg, Germany), a member of the Helmholtz Association HGF, for the provision of experimental facilities. Parts of this research were carried out at PETRA III. We want to acknowledge Chad Altobelli for advice on SPR assay design, and Leon Kraakman, Christin Radon, and Anja Drescher from Cytiva for advice on fitting SPR data.

## Author contributions

M.K. conceived the work, designed the compounds, performed the MS and TR-FRET experiments, and analyzed the data with contributions from C.O., L.C., C.G.N., and M.R.A. M.Z. and M.F. synthesized and characterized compounds. M.A.M.P. solved most of the crystal structures and performed the SPR experiments. J.M.V. performed the NanoBRET assay. J.L.R. was involved in the development and optimization of the TR-FRET assay. E.J.V. solved the initial crystal structures. M.K., C.G.N., M.R.A., C.O., and L.B. supervised the project. M.K. wrote the manuscript with contributions from all authors.

## Competing interests

Michelle R. Arkin, Christian Ottmann, and Luc Brunsveld are co-founders of Ambagon Therapeutics. M.K., M.Z., M.F., C.G.N,. and M.R.A. are co-inventors on a patent application related to this work. The remaining authors declare no competing interests.
