## [Transparent Peer Review file · Nature Communications]

Scaffold-hopping for molecular glues targeting the 14-3-3/ER α complex

Corresponding Author: Professor Michelle Arkin

Version 0:

Reviewer comments:

Reviewer #1

(Remarks to the Author)

Review of article: "Rapid scaffold-hopping for molecular glues: from fragments to cell-active probes targeting the 14-3-3/ER α complex" by Konstantinidou et al.

Noteworthy results

The claim of this study is to showcase the possibility of using a scaffold hopping approach from compound 127 through rapid derivatization (MCR) to improve the overall EC₅₀ of these compounds, as measured by BRET interaction in cells. These covalent warheads were elaborated with small modifications and assessed in TR-FRET & intact mass spec assays, and afterwards through the use of structural biology. Compounds do not seem to be designed using structure-based principles, but rationalized after-the-fact. It is not clear that this new scaffold is significantly better than the old scaffold, but is different. Data generation appears to be done in a robust, reproducible manner, but the interpretation of the modifications as impacting K_{inact} vs. K_i is not fully fleshed out.

Background

2 different series were identified and small changes affected the SAR:

Starting from structure of compound 127 bound to 14-3-3s/ER- α complex, the authors generated the alternative imidazo pyridine series (benzyl vs. phenyl analogues). Compounds were selected by mass spec with fast binding kinetics. Small ring substitutions generated SAR through the TR-FRET and intact Mass spec assays and then reconciled with crystal structures.

Most of the co-crystal structures were solved to adequate resolution to reveal atomic level detail for the compounds (1.5Å or better). Only the chloroacetamide warheads are shown in crystal structures. The only co-complex structure that is a little concerning is PDB ID 9I70 (compound 17) where the B-factor for the ligand is 67, which is much higher than the other structures. It seems that the density for compound 17 in 2Fo-Fc maps (Fig S11) is also most questionable in the linker region. Because this compound has the best BRET EC₅₀ (2.7 μ M) and has the highest reactivity of all the compounds in the mass spec assay (with ER α), it is one of the most interesting ones for structural interpretation.

The authors argue based on 2Fo-Fc densities (Fig. S11) that the amide linker shifts from Cis (cpd 21, 17) to trans (cpd 19, 41, 42, 40) but it's not clear what is driving this shift. Furthermore, this is the region of the density that is broken in most compounds. It is intriguing that the amide bond shifts from cis (cpd 17) to trans (cpd 41) as compounds are elaborated. Compound 17 is the most potent in BRET assay, but cpd 41 has the biggest FRET E_{max} (though compound 181 and FC-A have the largest FRET E_{max})

1) While it is clear that the density fits the structures in the biased 2Fo-Fc map (Fig S11), it would be more relevant to show the omit map for these compounds. If the amide bond is critical for the interpretation of the complementarity of the interactions within the pocket, then the unbiased density map should be included in supplemental data.

2) Furthermore, the surface complementarity between the ligand and its binding site should be addressed, through understanding the VdW or electrostatic complementarity of the compounds—to see if these correlate with the mass spec or EC₅₀ (BRET) assay results. (For example, ConCAVITY, PLB, MF-PLB, Ligsite-csc)

Electrostatic Complementarity as a Fast and Effective Tool to Optimize Binding and Selectivity of Protein–Ligand Complexes | Journal of Medicinal Chemistry

Evaluation by SPR to argue differences in K_{off} for compound 181 vs. compounds 17 and 41.

Figure 4 shows that although compound 181 was described earlier (Konstantinidou et al., 2023), in the modified SPR assay in which dissociation of the complex upon washing is used to measure K_ds, that there is a general trend of compounds showing slowest off-rate and their effect on cellular activity. However, the off-rate for compound 181 is still superior to the

compounds from this study (10 and 17).

Covalent mass spec assay (intact) to understand Kinact/Ki:

The binding measurement performed in the absence of phospho-ERalpha peptide was to demonstrate whether compound bound independently or was synergistically binding with ER-alpha peptide binding --a concentration-dependent time course was run in both states. However, this study does not provide any analysis of the binding affinity vs. the reactivity of warhead (Kinact/Ki) by analysis of their mass spec or TR-FRET data. Mechanism-Based Inhibition: Deriving KI and kinact Directly from Time-Dependent IC50 Values which might require shorter time frames than 1 hr (utilizing GSH to quench the experiment).

It is possible that all these substitutions do is improve the orientation of the warhead to enact a faster chemical bond, rather than enhance the complementarity of the rest of the compound towards its binding site.

Novelty of this work:

This scaffold-hopping study has built on prior work from Visser E.J et al (Angew Chem Int Ed, 2023), Konstantinidou et al, J. Am Chem Soc, 2023) and Guillory X. et al. (JMed Chem, 2020). The new approaches were to utilize a different starting scaffold and to rely on alternative assays to confirm cooperativity rather than utilizing a fluorescence anisotropy assay (FA), which was plagued by cy5 ERalpha-peptide dye-binding artifacts. The new TR-FRET assay described here utilizes a biotinylated- ERalpha peptide rather than a Cy5 peptide, and this also demonstrated difference from ITC determined Kds, due to the tetravalent nature of the streptavidin (acceptor). The mass spec assay (unlabelled) is viewed as a more robust assay as a result of the fact that there is no dye attached to the peptide.

These recent covalent compounds appear to stabilize in the low micromolar range while former compounds stabilized in the triple digit micromolar EC50 stabilization range. Thus, the novelty of this new scaffold is likely its rigidity and drug-likeness, which is likely leading to enhanced EC50s.

Recommendation- manuscript is not quite addressing the critical question: This work utilizes multiple biophysical techniques and generally demonstrates a chain of translation from invitro Mass spec to off-rate to cell-based activity; the scaffold-hopping approach is somewhat novel. But ultimately, these compound do not have a significantly enhanced EC50 relative to the original compound 181 (former series). The key question which is not really addressed in this analysis is whether these small modifications on the new scaffold are enhancing the binding affinity vs. enhancing the reactivity by properly positioning the warhead.

If the argument is that to enhance EC50 by reducing the off-rate, it would be valuable to know which modifications were enhancing the binding rather than reactivity of the covalent compound. The manuscript should be revised to include a Kinact/Ki analysis, correlate Ki with electrostatic complementarity of the ligand/binding site and propose how to enhance the off-rate further by enhancing the electrostatic and van der waals complementarity of the compounds.

Reviewer #2

(Remarks to the Author)

The manuscript by Konstantinidou and colleagues reports on the systematic development of new chemical matter that stabilizes the 14-3-3/ER interaction by a molecular glue-like mechanism. Towards this aim, the authors took a previously known covalent stabilizer of the 14-3-3/ER interaction as starting point and computationally derived a new chemically diverse scaffold that fulfills the same task. In the following, the authors utilized multiple biophysical assays supported by an impressive amount of co-crystal structures of ternary complexes to enable a sound SAR and rationally optimize the 14-3-3/ER stabilizers from neutral binders to cooperative stabilizers. Along the way, intact MS, TR-FRET and SPR assays are employed to monitor cooperativity and kinetics. The supporting co-crystal structures nicely explain difference in the binding mode of very close analogs that ultimately affect activity. Finally, the authors provide initial evidence in form of a nanoBRET dimerization assay that their best compounds are active in cells.

Overall, this is an impressive paper that addresses an important problem in the field of molecular glues – the development of new scaffolds that can glue proteins together. The manuscript is very well-written and the experiments are performed and interpreted appropriately. Especially, the computational approach is novel and interesting. Additionally, the soundly established SAR together with many co-crystal structures really helps to understand how minor modifications and influence the binding mode and ultimately the efficacy of the molecular glues. The manuscript should be of interest to the readers of Nat. Comm. and everyone in the molecular glue field.

Comments to the authors:

1. The authors put forth the claim of cell activity in the title: "Rapid scaffold-hopping for molecular glues: from fragments to cell-active probes targeting the 14-3-3/ERa complex". While the authors convincingly show in nanoBRET assays that their compounds indeed stabilize the 14-3-3/ERa interaction in cells, it remains an open question whether or not this has any functional consequences on ERa signaling. In my opinion, the paper would improve with some more biological data that support relevant in-cell activity.
2. While the ligand overlay of compound 127 and docking pose of the new MCR scaffold (Fig 1B) is helpful, showing the chemical structure of 127 next to the general MCR scaffold (Fig 1C) would be beneficial for direct comparison. Following the same idea, showing the structure of reference compound 181 in Fig. 4 would be great.
3. The supplementary figure S8 reveals that 181 and FCA have been tested and in the TR-FRET assays and appear to be strongest stabilizers within the set of compounds. It would be worthwhile showing the data in the main manuscript, including

FCA in the nanoBRET assays and discussing possible reasons between in-vitro and cell activity. Additionally, the authors state the number of replicates and show error bars for the TR-FRET measurement in the figure captions.

4. As a hub protein, 14-3-3 interacts with several clients. To confirm the selectivity of the new scaffolds an investigation of selectivity versus a panel of 14-3-3 clients would be worthwhile.

Reviewer #3

(Remarks to the Author)

It is my opinion as a reviewer that the science is of high quality and worthy of publication.

The authors present a novel strategy iterating on their current work identifying molecular glues for a model protein-protein interaction between 14-3-3 & a peptide fragment of ER α . The strategy utilizes previous structural biology data generated from by their research group and applies AnchorQueryTM to successfully identify a new covalent molecular glue for this protein-protein interaction (PPI). According to the text, the computational technique hinges on screening through a fragment-based drug design approach with the constraint of identifying fragments which are approachable by synthesis through multicomponent reaction chemistry (MCR). The authors disclose 42 new structures which they test through successive rounds of biochemical & structural biology assays to validate their effectiveness as molecules glues. During the process of disclosing the structures some attempts are made to draw structure activity relationships (SAR) for series of molecules within the 42 disclosed structures. The multitudes of assay data used to support their claims appear to be of high quality, however, as synthetic/medicinal chemist, vetting the quality of the technical points of each assay is outside of the scope of my expertise. Finally, the authors propose that their work represents a new strategy towards scaffolding hopping from pre-existing molecular glues which is accessible based on the application of user-friendly MCR chemistry. There are a few points, if clarified which would enhance the understanding of the of the paper and its utility to scientists in adjacent fields.

Generally, further elaboration on the design strategy would be helpful in the results section of the text. While it is obvious after reading the synthesis/SAR section that each of the 3 components of the MCR, were varied at points in the study, it's not clear from the outset of the study what guided the selection of each component beyond synthetic tractability or commercial availability. The list of computational suggestions was not referenced directly in the paper, and no supplemental table was found by this reviewer. Commenting on the design rational from computation to the laboratory I think is helpful for a reader to understand how to apply this workflow themselves. For example, were the calculations run iteratively and refined as the library was synthesized or were all 42 compounds suggested by AnchorQuery? While I don't believe the authors are in any obligation to disclose more structures involved in the process, I do think the novelty and utility of their workflow would be enhanced by explaining how they proceeded from design space to library synthesis.

Specific questions:

Clarification: "The benzyl analog 1, which lacked aryl ring substitutions, was a neutral binder. Introduction of electron withdrawing groups, such as halogens or nitro-groups in the o-, m- or p-position (compounds 2-9) reduced binding in the presence of ER α , even in the case of a small F-substitution." Why highlighting EWG here? No EDG were disclosed. Is there an implied argument about electronics or solely positional substitution? I think more examples, with electron donating properties, would be needed to make an electronics argument.

Clarification: "Remarkably, a Me-group in the o-position (13) led to the first molecular glue of the series; compound 13 showed binding in the presence of ER α and very low apo binding" Not being an expert in Molecular glues, the required delta in binding of apo vs. exp is not obvious on the bar graphs. It was unclear what level is achieved by 13, which was not hit by 11 in Fig 1 E & S4. The authors could broaden audience understanding by indicating on the figure quantitatively discriminating binders from glues. Could the graphs indicate the threshold for significance in the delta of ER α vs. Apo binding in the MS assay?

Clarification: "In agreement with our previous observation, the introduction of additional substitutions in the p-position (-Cl, -Me, -OH) significantly reduced binding (compounds 22-25), especially for the triple substituted analogs (double o- and ppositions). Use of previous without referencing compound within paper. Para substitution was previously mentioned in reference to previous as being another cited paper. In that paper, para substitution was beneficial. I think adding compound numbers here to clarify to the reader.

Clarification: "The formation of these interactions seemed to favorably restrict the axial rotation of the aniline-phenyl bond, locking its position in a highly complementary shape with the composite 14-3-3/ER α surface." My understanding is this statement is indicating bias towards a single binding mode in the context of protein being bound to the protein. Is restricted rotation of the 2,6-dimethyl predicted in the gas-phase via computation? I don't think you would anticipate atropisomerism for this molecule in the 2-chloro-6-methyl case. The 1H NMR for compounds 18-19 doesn't appear to imply any restricted rotation on the NMR timescale.

Clarification: As a reader I was strongly interested to learn more about the initial design decisions which informed the first 16 compounds tested.

"To test whether this scaffold was suitable for the development of 14-3-3 σ /ER α stabilizers, we synthesized a small set of derivatives varying only the isocyanide position, which according to our design was expected to interact with K122 of 14-3-3. We included benzyl and phenyl isocyanides with a diverse substitution pattern on the aryl ring. We maintained the covalent chloroacetamide warhead, based on our extensive investigation of electrophiles in our previous work.²¹"

I understand limiting the initial screen to validate the scaffold. However, I'm interested in computational data or prerequisite data led the authors to believe that in the context of a new imidazopyridine scaffold, that maintaining a covalent region of the molecule and solely starting by modifying the aniline/benzylamine would provide due diligence. While this ends up successful, later in the article, the authors state that the previous anchor, p-chlorophenyl, unable to make the previous key

interaction (K122 halogen bond). The structural motif was stated to be foundational to the design calculation in the pharmacophore in AnchoQuery. Do the authors have any comment on how strong of a prediction tool AnchorQuery was retrospectively for their molecules? Was the design calculation solely useful for identifying imidazopyridine as a scaffold for linking the covalent warhead & aniline?

As a reader, my first thought is that perhaps the original important element of the pharmacophore could have been accommodated with covalent warhead tethers with different geometry. Why was a linear hypothesis tested first in comparison to using a matrix-style data set? While testing multiple combinations increases the synthetic burden, MCR strategies are ideally tailored to tackle this problem by utilizing a combinatorial matrix of the reactants. The highlighting of the MCR route by the authors made this question more pertinent to this disclosure than a typical synthetic campaign. The benzylamine vs. aniline accomplishes this within the same region, but what if the methylene had been inserted into the Y component (aldehyde). What if a shorter aldehyde-derived linker was used? Based on compound 32, significantly later in the manuscript, extension of the simple phenyl linker seems to have been not well tolerated, but I'm interested in whether this was reflected in the original design work which prompted the scaffold hop.

Supplemental Document:

Synthetic methods are of the expected quality. Compounds appear to all be novel and have adequate characterization data. Compounds 8, 10, 12, 14, 15, 17, 18, 22, 26, 27, 30, & 32 did not have tabulated mass spectrometry data. If this is a requirement for the journal, this should be added.

Characterization data for compound 9 was not found in the list of tabulated spectroscopy values.

Graphical representations of the NMR spectrum for some compounds were not found in the supplemental information, even when tabulated. If it is a requirement of the journal, then it should be added prior to publication. If the compounds were known and matched literature precedents, then this should be indicated in the tabulated values section and cited.

Version 1:

Reviewer comments:

Reviewer #1

(Remarks to the Author)

Review of article: "Rapid scaffold-hopping for molecular glues: from fragments to cell-active probes targeting the 14-3-3/Erα complex" by Konstantinidou et al.

I thank the authors for addressing all of my comments and questions. The authors have addressed the comments and questions I had in a comprehensive way and the manuscript has been improved.

Upon review of the author's explanations, I feel the authors have addressed the concerns I had related to the novelty of this work. I agree that the lack of established methodologies for molecular glues and the focus on the kinetics of compound binding and the cooperativity of the protein-protein interaction makes this paper unique. I am also satisfied with the author's explanation for the reduced density around the linker by their inclusion of the omit maps in supplementary figure 13.

However, if the occupancy of the compound has been reduced to lower the B-factor in the refinement, the occupancy should be stated for each compound in crystallography tables in supplementary table 6 to clarify that the occupancy is a variable.

The addition of supplementary figure 7 does support the argument that the pocket is mainly hydrophobic and that shape complementarity may be critical.

The authors clearly show that the compounds which covalently react with the 14-3-3:ERα complex over 14-3-3 alone also show reduced off-rate by SPR. Adding supplementary figures 5 and 11, using the off-rates of compounds 181, 17 and 41 as a measure of thermodynamic specificity, the authors show a trend that suggests that the thermodynamic stabilization of the complex by the glue. This validates the use of the cooperative mass spec assay as a front-line SAR-driving assay.

The paper can be published as it stands, with minor corrections to include occupancy in the crystallography tables. The only remaining comment I have is that it was a bit surprising to me is that most of the end-state crystal structures that the authors have solved in the presence of the ERα peptide look very similar, despite significant differences in the compound's reactivity rates or overall stability. For example, compound 25, which may have been designed to enhance hydrogen bonding by adding a para hydroxyl, actually has reduced activity in the various assays. Does this suggest that it is difficult to design in specific polar contacts into a largely hydrophobic binding surface? A small nod to the lessons learned regarding structure-based drug design and its utility in predicting improved potency for molecular glues would enhance the paper's impact even further.

Reviewer #2

(Remarks to the Author)

I thank the authors for fully addressing all my concerns. It is exciting to hear that this publication will be backed up by a followup paper in due course!

I recommend publication of this manuscript.

Reviewer #3

(Remarks to the Author)

The revisions provided by the authors have satisfied my critiques & comments. Specifically, I found the added information regarding the use of AnchorQuery informative from the perspective of a medicinal chemist who would want to learn how to apply the author's strategy. Further, clarifications, with more specific references towards prior works, enabled me to have a much better understanding of how the authors chose the compounds which were disclosed. I also found the further clarification within the body of the text which explained data which suggests molecular glue vs. protein binding to enhance my understanding of the authors data interpretation. Considering the characterization data provided within the supplemental document, for library compounds, I think it is quite clear that the compounds proposed match the presented characterization data. I would still strongly encourage the authors to add to their supplemental stating a reason why MS data is not provided on the per compound basis. As this paper is eventually indexed by search engines and read by scientists potentially out of context or without the MS expertise possessed by the authors, I believe it to be good practice to include a reason when certain unobtainable data is not included.

The current version of the manuscript meets my requirements for publication.

REVIEWER COMMENTS

Reviewer #1 (Remarks to the Author):

Review of article: "Rapid scaffold-hopping for molecular glues: from fragments to cell-active probes targeting the 14-3-3/Era complex" by Konstantinidou et al.

Noteworthy results

The claim of this study is to showcase the possibility of using a scaffold hopping approach from compound 127 through rapid derivatization (MCR) to improve the overall EC50 of these compounds, as measured by BRET interaction in cells. These covalent warheads were elaborated with small modifications and assessed in TR-FRET & intact mass spec assays, and afterwards through the use of structural biology. Compounds do not seem to be designed using structure-based principles, but rationalized after-the-fact. It is not clear that this new scaffold is significantly better than the old scaffold, but is different. Data generation appears to be done in a robust, reproducible manner, but the interpretation of the modifications as impacting Kinact vs. Ki is not fully fleshed out.

Response: We thank reviewer 1 for taking the time to review our manuscript. We would like to clarify a few points.

The focus on the manuscript is to describe a rational discovery strategy for molecular glues, which remains a significant challenge in the field. We describe in detail our rationale for scaffold selection and structure-guided optimization based on the composite PPI surface and the numerous new crystal structures for our new MCR series. We have added more details for the design strategy also in response to comments by reviewer 3; we hope these additions clearly emphasize the structure-based principles used to design our compounds. (See below and revised manuscript).

The end goal of scaffold hopping is not necessarily to make a "better" scaffold, but to provide the rationale behind scaffold design, aiming to improve certain features, like rigidity in this case. Additionally, in most of our assays the new scaffold shows comparable (and slightly improved) activity to the previously disclosed series. We added a point in the Discussion to note that these compounds and FC-A come to a similar efficacy, perhaps due to the limited opportunity for interaction with the C-terminal valine.

One important aspect of this work is the focus on molecular glues; compounds that bind cooperatively in protein-protein interfaces. In contrast to inhibitors, these MGs bind to protein/protein interfaces, lacking enzymatic activity. Thus, traditional techniques for certain assays are not applicable, including calculations for Kinact vs Ki (which we routinely do for covalent inhibitors, see DOI: 10.1021/jacs.2c12240). We have thought about this issue at length. We have designed the mass spectrometry assay in such a way that allows the distinction of cooperative stabilizers versus neutral binders and provides kinetic information for binding in the form of a time-course (mass spectrometry data, Supplementary figures 2-3). Additionally, we also monitor how binding of the glue alters 14-3-3/peptide complex formation with the SPR experiment (figure 4f). Because we only see binding when compounds are covalent, and in the presence of the peptide, we lack an assay that can distinguish changes in kinact from Kd. In the text, we assert that the observed changes in cooperativity are much more likely due to compound/protein contacts than to chemical reactivity, given the very similar compound structures and the identical warhead orientations in the products observed by crystallography.

Background

2 different series were identified, and small changes affected the SAR.

Response: this manuscript describes on SAR only for the new MCR scaffold (which we consider one scaffold). We reference the paper for the original series (DOI: 10.1021/jacs.3c05161) and use only the best compound 181 as a positive control.

Starting from structure of compound 127 bound to 14-3-3s/ER-alpha complex, the authors generated the alternative imidazo pyridine series (benzyl vs. phenyl analogues). Compounds were selected compounds by mass spec with fast binding kinetics. Small ring substitutions generated SAR through the TR-FRET and intact Mass spec assays and then reconciled with crystal structures.

Most of the co-crystal structures were solved to adequate resolution to reveal atomic level detail for the compounds (1.5Å or better). Only the chloroacetamide warheads are shown in crystal structures.

Response: This comment is not clear to us. We present (and describe in detail) the crystal structures of the ternary complexes of the full compounds with 14-3-3/ER α phospho-peptide. Most of prepared compounds were chloroacetamides. All crystal structures showed compounds with chloroacetamide warheads.

The only co-complex structures that is a little concerning is PDB ID 9I70 (compound 17) where the B-factor for the ligand is 67, which is much higher than the other structures. It seems that the density for compound 17 in 2Fo-Fc maps (Fig S11) is also most questionable in the linker region. Because this compound has the best BRET EC50 (2.7 μ M) and has the highest reactivity of all the compounds in the mass spec assay (with ER α), it is one of the most interesting ones for structural interpretation.

The authors argue based on 2Fo-Fc densities (Fig. S11) that the amide linker shifts from Cis (cpd 21, 17) to trans (cpd 19, 41, 42, 40) but it's not clear what is driving this shift. Furthermore, this is the region of the density that is broken in most compounds. It is intriguing that the amide bond shifts from cis (cpd 17) to trans (cpd 41) as compounds are elaborated. Compound 17 is the most potent in BRET assay, but cpd 41 has the biggest FRET Emax (though compound 181 and FC-A have the largest FRET Emax)

Response: We agree that the B-factor of compound 17 was relatively high, therefore we performed refinements to derive the occupancy of the ligand. The occupancy was set at 0.81 after refinement, which resulted in a lower B-factor (42 instead of 67). The new pdb and mtz files are updated in the deposition and the SI table is updated.

Indeed, it is interesting that we observed different orientations in the amide bond for certain compounds, which we described in the text. We hypothesize that the positioning of the compounds, driven by interactions in close proximity to the client peptide, might affect the positioning of the warhead; however, we do not attempt to make strong statements. While the S-C bond shows the weakest density/is broken in several of the structures, the carbonyl density is strong, supporting the position of the linker in the model.

As the reviewer highlights, validating molecular glues is significantly more complex than inhibitors. We have noticed differences both in EC50s and Emax in different assays and we believe that reporting both numbers is more precise than choosing one over the other. Additionally, we validate the compounds in multiple assay formats and the best ones show consistent improvements across different readouts.

1) While it is clear that the density fits the structures in the biased 2Fo-Fc map (Fig S11), it would be more relevant to show the omit map for these compounds. If the amide bond is critical for the interpretation of the complementarity of the interactions within the pocket, then the unbiased density map should be included in supplemental data.

Response: the omit maps of all compounds were added in Supplementary Figure 13, clearly showing the density of the amide bonds.

2) Furthermore, the surface complementarity between the ligand and its binding site should be addressed, through understanding the VdW or electrostatic complementarity of the compounds—to see if these correlate with the mass spec or EC50 (BRET) assay results. (For example, ConCAVITY, PLB, MF-PLB, Ligsite-csc) Electrostatic Complementarity as a Fast and Effective Tool to Optimize Binding and Selectivity of Protein–Ligand Complexes | Journal of Medicinal Chemistry

Response: we read carefully the suggested publication (DOI: 10.1021/acs.jmedchem.8b01925), which refers to calculations and visualization of electrostatic complementarity of protein-ligand complexes. The composite surface in the 14-3-3/ER α complex, especially in proximity to ER α , is largely hydrophobic, with a very small contribution of electrostatic interactions. Hence, most of the observed interactions between our molecular glues and the complex are hydrophobic, and the calculation of electrostatic complementarity seems less relevant. As mentioned in the conclusion of the suggested publication, "It is worth noting that EC analysis alone is not going to provide a complete prediction of the binding free energies of protein–ligand complexes because of missing key factors such as conformational contributions for both proteins and ligands, desolvation effects, van der Waals (vdW) and entropic contributions, and energetics of the binding site water molecules." Several of the factors that are not part of the calculations are significant for the binding of molecular glues.

Taking the reviewer's request more conceptually, we included a new figure (Supplementary Fig 7). This figure illustrates the atom-colored surfaces of 14-3-3, ER α , and compound 41, emphasizing the hydrophobic interactions at the PPI interface.

Evaluation by SPR to argue differences in K_{off} for compound 181 vs. compounds 17 and 41.

Figure 4 shows that although compound 181 was described earlier (Konstantinidou et al., 2023), in the modified SPR assay in which dissociation of the complex upon washing is used to measure K_{ds} , that there is a general trend of compounds showing slowest off-rate and their effect on cellular activity. However, the off-rate for compound 181 is still superior to the compounds from this study (10 and 17).

Response: It is not clear to us if this comment is a suggestion from the reviewer or a point to clarify. We describe the assay in the main text; in summary, we pre-bind the covalent compound to 14-3-3, then immobilize the 14-3-3-compound conjugate to the SPR surface and measure the binding of the peptide (not the compound, which is covalently bound to 14-3-3). Nevertheless, the difference in off-rate for the ER α peptide in the presence of 181 is 0.016 1/s vs 0.021 1/s for 41, ie, almost identical.

This is the first report of developing an SPR assay for covalent molecular glues, so may be challenging to readers; hence, we sought to describe it in some detail. The key point was to study how the 14-3-3/peptide affinity and kinetics ($K_d = k_{on}/k_{off}$) were affected by the molecular glue. For this PPI, we saw that the compounds dramatically decreased the k_{off} . The compounds measured in the SPR strongly stabilized the 14-3-3/ER α complex (K_d values in range of 4 – 15 nM, approximately 100-200-fold lower than the 14-3-3/ER α in the absence of the glue, $K_d = 1 \mu\text{M}$). We added Supplementary Table 4 with SPR data and fitted parameters.

Covalent mass spec assay (intact) to understand K_{inact}/K_i :

The binding measurement performed in the absence of phospho-ER α peptide was to demonstrate whether compound bound independently or was synergistically binding with ER- α peptide binding --a concentration-dependent time course was run in both states. However, this study does not provide any analysis of the binding affinity vs. the reactivity of warhead (K_{inact}/K_i) by analysis of their mass spec or TR-FRET data. Mechanism-Based Inhibition: Deriving K_i and k_{inact} Directly from Time-Dependent IC_{50} Values which might require shorter time frames than 1 hr (utilizing GSH to quench the experiment).

Response: Since these compounds are molecular glues, the terminology must be changed, eg, not IC_{50} but EC_{50} , and not enzymatic activity but stoichiometric binding. As a minor point, the compounds are not quenched by GSH (at least at the millimolar concentrations in cells). Our intact mass spectrometry can measure how fast the compounds bind in the absence vs presence of ER α to a time resolution of minutes, which is adequate for these warheads. Indeed, as the reviewer notes, we use MS to distinguish neutral binders from molecular glues. As briefly mentioned in our response to 'noteworthy results,' we now measure % bound for the covalent compounds as a function of time. Since our compounds only bind in the presence of the ER α peptide, and the peptide binds rapidly to 14-3-3 (see Fig 4), we believe this is all we can measure and reasonably interpret.

Since studies for non-degradative molecular glues are relatively new (partially because of the lack of methods to rationally identify new chemical matter), there are new challenges on how to validate them. We believe that the biophysical studies provided here, with the requested amendments, advance the state-of-the-art and will be more widely used as the field of covalent PPI inhibitors and molecular glues continue to develop.

It is possible that all these substitutions do is improve the orientation of the warhead to enact a faster chemical bond, rather than enhance the complementarity of the rest of the compound towards its binding site.

Response: This hypothesis is interesting but is not supported by the data. We have added additional explanations in the main text. 1) The warheads and linkage to the scaffold are identical for most compounds described, independent of their stabilizing activity. 2) If we compare the crystal structures of compounds with the same warhead, for example 17 and 25 (Fig 2), we notice a perfect alignment of the two compounds, including the positioning of the warhead. The different interactions based on the compounds' substitution pattern are affecting their activity, as described in detail. Even though the warhead orientation is the same, compound 17 binds very fast in the mass spec assay, whereas 25 barely binds. 3) in the SPR assay, where compound is fully bound before the

measurement, peptide/protein Kds are shifted by over 100-fold, demonstrating an effective stabilization of the interface. We therefore conclude that the speed of binding is due to cooperativity, rather than the converse, as suggested.

Novelty of this work:

This scaffold-hopping study has built on prior work from Visser E.J et al (Angew Chem Int Ed, 2023), Konstantinidou et al, J. Am Chem Soc, 2023) and Guillory X. et al. (JMed Chem, 2020). The new approaches were to utilize a different starting scaffold and to rely on alternative assays to confirm cooperativity rather than utilizing a fluorescence anisotropy assay (FA), which was plagued by cy5 ERalpha-peptide dye-binding artifacts. The new TR-FRET assay described here utilizes a biotinylated- ERalpha peptide rather than a Cy5 peptide, and this also demonstrated difference from ITC determined Kds, due to the tetravalent nature of the streptavidin (acceptor). The mass spec assay (unlabelled) is viewed as a more robust assay as a result of the fact that there is no dye attached to the peptide.

Response: although this looks like a summary and not a question to address, we would like to clarify that the MS assay measures compound binding, not peptide binding, and does not measure stabilization of the protein-peptide interaction.

We usually use FP to measure the effect of the compound on the complex Kd. Here, the TR-FRET assay was used to circumvent the issue of scaffold fluorescence, which Cy5-Fp was not able to address. We decided to explain our assay design process because a) compound fluorescence on FITC-wavelengths is not uncommon and b) since the cy5 peptide led to non-specific binding, we aimed to emphasize an issue that sometimes gets overlooked. Therefore, we believe that our approach to solving these issues would be useful to the community.

These recent covalent compounds appear to stabilize in the low micromolar range while former compounds stabilized in the triple digit micromolar EC50 stabilization range.

Response: to clarify, the covalent compounds published in (DOI: 10.1021/jacs.3c05161) show EC50 values in the low micromolar range. Non-covalent molecules derived from fragments (doi.org/10.1002/anie.202308004) show 100-ish μ M EC50 values.

Thus, the novelty of this new scaffold is likely its rigidity and drug-likeness, which is likely lending to enhanced EC50s.

Recommendation- manuscript is not quite addressing the critical question: This work utilizes multiple biophysical techniques and generally demonstrates a chain of translation from invitro Mass spec to off-rate to cell-based activity; the scaffold-hopping approach is somewhat novel. But ultimately, these compounds do not have a significantly enhanced EC50 relative to the original compound 181 (former series). The key question which is not really addressed in this analysis is whether these small modifications on the new scaffold are enhancing the binding affinity vs. enhancing the reactivity by properly positioning the warhead.

Response: to us, the primary novelty of the manuscript is the approach to designing new molecular glues that bear low structural similarity to the starting points. Design of molecular glues is of high interest and low success in the literature to date. Thus, the main focus of the manuscript is to describe a novel, rational, and successful hopping approach for molecular glues. Since this is not a trivial task it is noteworthy that we already show the new MCR analogs have comparable activity to our previous series. As the reviewer notes, the new scaffold also has improved rigidity and is more-drug like.

If the argument is that to enhance EC50 by reducing the off-rate, it would be valuable to know which modifications were enhancing the binding rather than reactivity of the covalent compound. The manuscript should be revised to include a Kinact/Ki analysis, correlate Ki with electrostatic complementarity of the ligand/binding site and propose how to enhance the off-rate further by enhancing the electrostatic and van der waals complementarity of the compounds.

Response: as mentioned above, our point is not to focus only on one aspect of the binding, but instead to present a complete workflow on how to systematically validate novel molecular glues in carefully designed biophysical assays. We have already responded to the individual points and would like to state once again, that molecular glues require different metrics than inhibitors.

To summarize our findings, we show that the compounds that bind the fastest and at the lowest concentrations in the presence of the peptide also show the largest stabilization by FRET and NanoBRET. Stabilization is further validated and explained by 100+fold reductions in K_d for the protein-peptide interaction, driven mostly by reduced off-rates. Structural studies indicate minor changes in the shapes of the molecules but large changes in stabilization that correlate closely with improved interactions with the protein, as thoroughly explained with the co-crystal structures and SAR development with the mass spec data.

Reviewer #2 (Remarks to the Author):

The manuscript by Konstantinidou and colleagues reports on the systematic development of new chemical matter that stabilizes the 14-3-3/ER α interaction by a molecular glue-like mechanism. Towards this aim, the authors took a previously known covalent stabilizer of the 14-3-3/ER interaction as starting point and computationally derived a new chemically diverse scaffold that fulfills the same task. In the following, the authors utilized multiple biophysical assays supported by an impressive amount of co-crystal structures of ternary complexes to enable a sound SAR and rationally optimize the 14-3-3/ER stabilizers from neutral binders to cooperative stabilizers. Along the way, intact MS, TR-FRET and SPR assays are employed to monitor cooperativity and kinetics. The supporting co-crystal structures nicely explain difference in the binding mode of very close analogs that ultimately affect activity. Finally, the authors provide initial evidence in form of a nanoBRET dimerization assay that their best compounds are active in cells.

Overall, this is an impressive paper that addresses an important problem in the field of molecular glues – the development of new scaffolds that can glue proteins together. The manuscript is very well-written and the experiments are performed and interpreted appropriately. Especially, the computational approach is novel and interesting. Additionally, the soundly established SAR together with many co-crystal structures really helps to understand how minor modifications and influence the binding mode and ultimately the efficacy of the molecular glues. The manuscript should be of interest to the readers of Nat. Comm. and everyone in the molecular glue field. *Response: We thank reviewer 2 for appreciating our work and highlighting the key challenge we are addressing in this work; the rational development of novel molecular glue scaffolds.*

Comments to the authors:

1. The authors put forth the claim of cell activity in the title: “Rapid scaffold-hopping for molecular glues: from fragments to cell-active probes targeting the 14-3-3/ER α complex”. While the authors convincingly show in nanoBRET assays that their compounds indeed stabilize the 14-3-3/ER α interaction in cells, it remains an open question whether or not this has any functional consequences on ER α signaling. In my opinion, the paper would improve with some more biological data that support relevant in-cell activity.

Response: We understand the point raised by the reviewer regarding the title. Although we provide limited cellular data in this paper, a more extensive biological evaluation of these compounds will be reported in due course (as a separate paper, due to the amount of data). Thus, we propose to change the title to: Rapid scaffold-hopping for molecular glues targeting the 14-3-3/ER α complex

2. While the ligand overlay of compound 127 and docking pose of the new MCR scaffold (Fig 1B) is helpful, showing the chemical structure of 127 next to the general MCR scaffold (Fig 1C) would be beneficial for direct comparison. Following the same idea, showing the structure of reference compound 181 in Fig. 4 would be great.

Response: we thank reviewer 2 for the suggestion. We have included the chemical structure of 127 in figure 1. In figure 4, we added the chemical structures of compounds 181, 17, 41 (used in SPR) and 85 (used in the NanoBRET assay as a negative control).

3. The supplementary figure S8 reveals that 181 and FCA have been tested and in the TR-FRET assays and appear to be strongest stabilizers within the set of compounds. It would be worthwhile showing the data in the main manuscript, including FCA in the nanoBRET assays and discussing possible reasons between in-vitro and cell activity. Additionally, the authors state the number of replicates and show error bars for the TR-FRET measurement in the figure captions.

Response: for the TR-FRET assay, as mentioned in the methods section, at least two independent experiments were performed for each compound. We have added this sentence in the caption of figure 4. We updated fig 4 and fig S9 to include the error bars for the TR-FRET assay.

We agree with the reviewer's implied comment that there are several parameters to consider when declaring one series 'the best.' Although our assay controls 181 and FC-A showed significant fold increase (high E_{max} value), the AppK_d values were weaker than the most potent MCR compound (41) (Table S3). So, regarding fold-stabilization compound 41 was more potent than 181 and FC-A. This assay is also challenging to quantify due to the hook effect. We would therefore prefer to keep these data in fig S9 and only discuss them in the text to avoid distracting the reader from the main story, which is the validation of the MCR scaffold.

In the more informative cell-based NanoBRET assay – which lacks the hook effect – compound 181 showed weaker stabilization than 41, comparing the EC₅₀ and fold-increase values (Table S4). FC-A was tested in this assay as a positive control, however at 30 μM showed a modest increase in the BRET signal. We have added this NanoBRET data in fig 4G. Additionally, since FC-A is non-covalent, the observed effect is not specific to the 14-3-3 sigma isoform. Thus, we have updated fig S11 with the FC-A data, both with 14-3-3 sigma and the 14-3-3^{C38N} mutant.

4. As a hub protein, 14-3-3 interacts with several clients. To confirm the selectivity of the new scaffolds an investigation of selectivity versus a panel of 14-3-3 clients would be worthwhile.

Response: we understand that the issue of selectivity is important for hub proteins such as 14-3-3. In our previous work, the optimization of compound 181 (DOI: 10.1021/jacs.3c05161) we tested the compound selectivity in a small selectivity panel with 8 different clients. Our current platform to investigate selectivity relies on FITC-labeled phospho-peptides. Unfortunately, this was the limiting factor for testing the selectivity of the new MCR compounds in the same assay. The scaffold was fluorescent in the FITC-wavelengths, which was the main reason for the development of the 14-3-3/ERα TR-FRET assay with the biotin-peptide. At the moment, we haven't developed TR-FRET assays for other 14-3-3 clients. Future cell-based studies will be more suitable for selectivity measurements; hence, we have not emphasized the selectivity of the MCR scaffolds in this manuscript.

Reviewer #3 (Remarks to the Author):

It is my opinion as a reviewer that the science is of high quality and worthy of publication.

The authors present a novel strategy iterating on their current work identifying molecular glues for a model protein-protein interaction between 14-3-3 & a peptide fragment of ERα. The strategy utilizes previous structural biology data generated from by their research group and applies AnchorQuery™ to successfully identify a new covalent molecular glue for this protein-protein interaction (PPI). According to the text, the computational technique hinges on screening through a fragment-based drug design approach with the constraint of identifying fragments which are approachable by synthesis through multicomponent reaction chemistry (MCR). The authors disclose 42 new structures which they test through successive rounds of biochemical & structural biology assays to validate their effectiveness as molecules glues. During the process of disclosing the structures some attempts are made to draw structure activity relationships (SAR) for series of molecules within the 42 disclosed structures. The multitudes of assay data used to support their claims appear to be of high quality, however, as synthetic/medicinal chemist, vetting the quality of the technical points of each assay is outside of the scope of my expertise. Finally, the authors propose that their work represents a new strategy towards scaffolding hopping from pre-existing molecular glues which is accessible based on the application of user-friendly MCR chemistry. There are a few points, if clarified which would enhance the understanding of the of the paper and its utility to scientists in adjacent fields.

Response: We thank reviewer 3 for positive remarks and recognition of the significance of our work.

Generally, further elaboration on the design strategy would be helpful in the results section of the text. While it is obvious after reading the synthesis/SAR section that each of the 3 components of the MCR, were varied at points in the study, it's not clear from the outset of the study what guided the selection of each component beyond synthetic tractability or commercial availability. The list of computational suggestions was not referenced directly in the paper, and no supplemental table was found by this reviewer. Commenting on the design rational from

computation to the laboratory I think is helpful for a reader to understand how to apply this workflow themselves. For example, were the calculations run iteratively and refined as the library was synthesized or were all 42 compounds suggested by AnchorQuery? While I don't believe the authors are in any obligation to disclose more structures involved in the process, I do think the novelty and utility of their workflow would be enhanced by explaining how they proceeded from design space to library synthesis.

Response: We understand that our readers might have limited experience with certain aspects of computational designs.

In the Results section (structure activity relationships), paragraphs 2 and 3 explained how AnchorQuery works. We have now added clarifications in this section for the points raised by the reviewer. In summary, AnchorQuery provides multiple possible hits per query, which are ranked according to energy minimization or RMSD. In contrast to other computational software, the initial list of computational suggestions is shown on the AnchorQuery's website at the time that the queries are submitted but is not stored permanently. We used the RMSD fit to select compounds having similar 3D shape as compound 127, such as the docking hit shown in figure 1B. Then on this hit, we performed structure-guided optimization to improve interactions with 14-3-3 and ER α . AnchorQuery did not provide a list of the 42 compounds that we synthesized, only the initial scaffold selection. We performed different rounds of scaffold optimization with the docking software SeeSAR, as mentioned in the 4th paragraph of the SAR section. We also added additional explanations there for clarification.

Regarding building block selection, we added a clarification for the approach, highlighted in the text. In summary, synthetic tractability or commercial availability is an advantage of MCR chemistry, but in our case the choice of building blocks was based on specific ligand-protein interactions we were aiming to improve. Each MCR building block selection focused on specific interactions either with 14-3-3 or with ER α . In short, the initial scaffold selection was guided by the AnchorQuery software. The evolution of compounds was primarily ordered by the multitude of co-crystal structures that we obtained. Therefore, we have performed very rational changes to the GBB scaffold, i.e. the transition from benzyl isocyanides to phenyl ones to improve cooperative binding at the PPI interface.

Specific questions:

Clarification: "The benzyl analog 1, which lacked aryl ring substitutions, was a neutral binder. Introduction of electron withdrawing groups, such as halogens or nitro-groups in the o-, m- or p-position (compounds 2-9) reduced binding in the presence of ER α , even in the case of a small F-substitution." Why highlighting EWG here? No EDG were disclosed. Is there an implied argument about electronics or solely positional substitution? I think more examples, with electron donating properties, would be needed to make an electronics argument.

Response: We added explanations in the main text. In short, the initial selection of EWG groups here was based on previous knowledge from fragment screens and the optimization of the 181 series (DOI: 10.1021/jacs.3c05161). Since at that time we were trying to validate the scaffold overall, we limited the substituent selection to EWG and decided to test their effect on different positions on the aryl ring. Later on, starting from analog 13, we include EDG groups.

Clarification: "Remarkably, a Me-group in the o-position (13) led to the first molecular glue of the series; compound 13 showed binding in the presence of ER α and very low apo binding" Not being an expert in Molecular glues, the required delta in binding of apo vs. exp is not obvious on the bar graphs. It was unclear what level is achieved by 13, which was not hit by 11 in Fig 1 E & S4. The authors could broaden audience understanding by indicating on the figure quantitatively discriminating binders from glues. Could the graphs indicate the threshold for significance in the delta of ER α vs. Apo binding in the MS assay?

Response: we understand the point raised by the reviewer. Although we would like to be able to set a threshold to quantify differences in binding between neutral binders and stabilizers, in our experience this is difficult to generalize. In addition to the figures, table S2 includes the % bound in the apo and ER α experiment. To clarify this statement, we added the % bound numbers for compounds 1, 10, 11 and 13 in the main text:

Compound 13 at 1 μ M showed 0% binding (1 hr measurement) and only 2.9% (24h), whereas binding in the presence of ER α increased from 25% (1hr) to 86% (24h). In other words, the binding was almost entirely driven by the presence of ER α . For compound 11, the difference between apo binding and ER α binding is smaller, and the

overall binding observed in the presence of ER α is roughly half of the one observed for compound 11. The structural differences between the compound (o-Me for 13 vs p-Cl for 11) made us realize that the o-substitution in this case correlated more with cooperative binding than the p-substitution, therefore we focused on analogs of compound 13.

Clarification: "In agreement with our previous observation, the introduction of additional substitutions in the p-position (-Cl, -Me, -OH) significantly reduced binding (compounds 22-25), especially for the triple substituted analogs (double o- and positions). Use of previous without referencing compound within paper. Para substitution was previously mentioned in reference to previous as being another cited paper. In that paper, para substitution was beneficial. I think adding compound numbers here to clarify to the reader.

Response: we rephased this sentence and added the compound numbers to clarify.

Clarification: "The formation of these interactions seemed to favorably restrict the axial rotation of the aniline-phenyl bond, locking its position in a highly complementary shape with the composite 14-3-3/ER α surface." My understanding is this statement is indicating bias towards a single binding mode in the context of protein being bound to the protein. Is restricted rotation of the 2,6-dimethyl predicted in the gas-phase via computation? I don't think you would anticipate atropisomerism for this molecule in the 2-chloro-6-methyl case. The 1H NMR for compounds 18-19 doesn't appear to imply any restricted rotation on the NMR timescale.

Response: The reviewer is correct, and we have added text to clarify that we find the protein selects to bind to one conformation. Our NMR data for these compounds do not imply restricted rotation. Our assumption is that most of these analogs could be classified as class-1 atropoisomers, which are commonly treated as achiral. From a chemical perspective, the rotational bond in these cases can sample the full 360 degrees of rotational conformations about the axis. From a biological perspective, they are typically binding to a given target in only a subset of these conformations. This varies depending on the surrounding amino acids. For example, comparing the crystal structures of analogs 19 and 21 shows the two extreme positions of the double ortho-substituents, driven by the favorable interactions and steric effects with 14-3-3. We added these explanations in the main text.

For the reviewer's interest, a recent publication by Neochoritis et al (<https://doi.org/10.1002/ejoc.202500212>) investigated atropisomerism for polysubstituted analogs in the context of the Groebke-Blackburn-Bienaymé reaction from a chemical perspective.

Clarification: As a reader I was strongly interested to learn more about the initial design decisions which informed the first 16 compounds tested.

"To test whether this scaffold was suitable for the development of 14-3-3 σ /ER α stabilizers, we synthesized a small set of derivatives varying only the isocyanide position, which according to our design was expected to interact with K122 of 14-3-3. We included benzyl and phenyl isocyanides with a diverse substitution pattern on the aryl ring. We maintained the covalent chloroacetamide warhead, based on our extensive investigation of electrophiles in our previous work.²¹"

Response: we have added additional explanations in response to the earlier comment regarding EWD groups in benzyl and aniline analogs and selection of building blocks. The decisions were made according to the structural information we had at the time. The updated text states:

*"Since ER α is phosphorylated on the penultimate residue, the binding interface to 14-3-3 creates a large, solvent-exposed pocket that can accommodate a small molecule. In general, 14-3-3 molecular glues are expected to stabilize 14-3-3/client complexes based on molecular recognition and shape complementarity. For the 14-3-3/ER α complex, the interactions with the terminal V595 of ER α , which is located close to the small pocket formed around K122 of 14-3-3 is the main part of the interface that needs to be targeted, as shown in Fig 1A for **127**. Thus, for the MCR scaffold our first modifications focused on that part of the interface." And later, "The size of the pocket allows the presence of aromatic rings and since surrounding 14-3-3 amino acids were primarily hydrophobic, we chose appropriate hydrophobic substituents."*

I understand limiting the initial screen to validate the scaffold. However, I'm interested in computational data or prerequisite data led the authors to believe that in the context of a new imidazopyridine scaffold, that maintaining a covalent region of the molecule and solely starting by modifying the aniline/benzylamine would provide due

diligence. While this ends up successful, later in the article, the authors state that the previous anchor, p-chlorophenyl, unable to make the previous key interaction (K122 halogen bond). The structural motif was stated to be foundational to the design calculation in the pharmacophore in AnchoQuery. Do the authors have any comment on how strong of a prediction tool AnchorQuery was retrospectively for their molecules? Was the design calculation solely useful for identifying imidazopyridine as a scaffold for linking the covalent warhead & aniline?

Response: We believe we have addressed this comment through the requested edits described above. To summarize, 1) the covalent bond is important from a structural perspective. We were developing molecular glues able to bind to the 14-3-3 sigma isoform, which has a cysteine in position 38. This cysteine is unique to the sigma isoform, potentially allowing selectivity over other isoforms. As mentioned in the introduction, our previous works focused on fragment-based screens for this specific residue ("We used a site-directed fragment-based technology, termed "disulfide tethering" with intact mass spectrometry as the readout to identify reversible fragments bound at the native cysteine (C38) of 14-3-3σ in the presence of a phosphorylated peptide that represented the disordered C-terminus of ERα"). 2) We started by modifying the aniline/benzylamine positions first, due to its proximity to the unique 14-3-3/ERα interface. The covalent warhead is at the rim of the interface, far from the client. 3) AnchorQuery is not meant for covalent docking. We used it as a starting point for scaffold modification and as explained in detail, performed structure-guided optimization based on SAR and crystal structures. 4) We have clarified why the presence of the halogen bond with K122 for these series did not correlate with improved cooperativity (paragraph comparing the crystal structures of compounds 17-25 and how their substitutions affect the axial rotation in the K122 pocket).

As a reader, my first thought is that perhaps the original important element of the pharmacophore could have been accommodated with covalent warhead tethers with different geometry. Why was a linear hypothesis tested first in comparison to using a matrix-style data set? While testing multiple combinations increases the synthetic burden, MCR strategies are ideally tailored to tackle this problem by utilizing a combinatorial matrix of the reactants. The highlighting of the MCR route by the authors made this question more pertinent to this disclosure than a typical synthetic campaign. The benzylamine vs. aniline accomplishes this within the same region, but what if the methylene had been inserted into the Y component (aldehyde). What if a shorter aldehyde-derived linker was used? Based on compound 32, significantly later in the manuscript, extension of the simple phenyl linker seems to have been not well tolerated, but I'm interested in whether this was reflected in the original design work which prompted the scaffold hop.

Response: We hope that these questions are addressed in the revised manuscript, where we seek to highlight our plan to scaffold-hop in the context of the covalent warhead. In our previous work (DOI: 10.1021/jacs.3c05161) we have tested multiple covalent warheads in a matrix-style. In that manuscript, we described in detail their differences in kinetics, binding and cooperativity, and based on these studies we maintained the optimal chloroacetamide warhead in this manuscript. We tested additional warheads in this work (34-38) and noticed that most of them were inactive or chemically unstable, supporting our choice of focusing on the chloroacetamide. While it may be true that remodeling the glue from scratch, testing multiple warheads and scaffolds in a matrix, our goal was not to describe a combinatorial chemistry approach but rather a rational design approach. We therefore optimized the interactions in the K122 pocket, then moved towards modifying other positions of the compound to establish a solid SAR.

Supplemental Document:

Synthetic methods are of the expected quality. Compounds appear to all be novel and have adequate characterization data.

Compounds 8, 10, 12, 14, 15, 17, 18,22,26,27, 30, &32 did not have tabulated mass spectrometry data. If this is a requirement for the journal, this should be added.

Characterization data for compound 9 was not found in the list of tabulated spectroscopy values.

Graphical representations of the NMR spectrum for some compounds were not found in the supplemental information, even when tabulated. If it is a requirement of the journal, then it should be added prior to publication. If the compounds were known and matched literature precedents, then this should be indicated in the tabulated values section and cited.

Response: our understanding regarding requirements for Nature Communications is that "combinatorial libraries should include standard characterization data for a diverse panel of library components". Here we describe SAR for

chemically similar analogs, based on a common MCR reaction; hence, it is not a requirement to provide HRMS for all compounds. Rather, we have provided mass spectrometry data, NMR characterization and included representative examples of NMR spectra. Together with the significant number of co-crystal structures, in our opinion, the compounds' characterization data is sufficient for this type of study.

However, we were able to obtain HRMS data for more compounds (8, 10, 12, 14, 17, 18, 19, 22, 26, 27, 28, 32), so we included them in the revision. Compounds 15 and 30 were not available for HRMS, but we had already provided the NMR spectra. These two compounds were weak in the biophysical assays and not the key examples for SAR. Regarding the NMR data for compound 9, which was an inactive analog early in the SAR series, adding NMR data does not seem like a valuable addition. Since the analog had a nitro group, MS data were not robust enough, a common issue with ionization of nitro-containing compounds.

REVIEWERS' COMMENTS

Reviewer #1 (Remarks to the Author)

Review of article: "Rapid scaffold-hopping for molecular glues: from fragments to cell-active probes targeting the 14-3-3/Erα complex" by Konstantinidou et al.

I thank the authors for addressing all of my comments and questions. The authors have addressed the comments and questions I had in a comprehensive way and the manuscript has been improved. Upon review of the author's explanations, I feel the authors have addressed the concerns I had related to the novelty of this work. I agree that the lack of established methodologies for molecular glues and the focus on the kinetics of compound binding and the cooperativity of the protein-protein interaction makes this paper unique. I am also satisfied with the author's explanation for the reduced density around the linker by their inclusion of the omit maps in supplementary figure 13. However, if the occupancy of the compound has been reduced to lower the B-factor in the refinement, the occupancy should be stated for each compound in crystallography tables in supplementary table 6 to clarify that the occupancy is a variable.

The addition of supplementary figure 7 does support the argument that the pocket is mainly hydrophobic and that shape complementarity may be critical.

The authors clearly show that the compounds which covalently react with the 14-3-3:ERα complex over 14-3-3 alone also show reduced off-rate by SPR. Adding supplementary figures 5 and 11, using the off-rates of compounds 181, 17 and 41 as a measure of thermodynamic specificity, the authors show a trend that suggests that the thermodynamic stabilization of the complex by the glue. This validates the use of the cooperative mass spec assay as a front-line SAR-driving assay.

The paper can be published as it stands, with minor corrections to include occupancy in the crystallography tables. The only remaining comment I have is that it was a bit surprising to me is that most of the end-state crystal structures that the authors have solved in the presence of the ERα peptide look very similar, despite significant differences in the compound's reactivity rates or overall stability. For example, compound 25, which may have been designed to enhance hydrogen bonding by adding a para hydroxyl, actually has reduced activity in the various assays. Does this suggest that it is difficult to design in specific polar contacts into a largely hydrophobic binding surface? A small nod to the lessons learned regarding structure-based drug design and its utility in predicting improved potency for molecular glues would enhance the paper's impact even further.

Response: We thank the reviewer for the positive comments. We added the compound occupancy numbers in supplementary table 6 (yellow highlights). For all compounds the occupancy was 1, apart from compound 17 (0.81).

We added the sentence “Since the 14-3-3/ER α interface was largely hydrophobic, most of the favorable ligand modifications were non-polar, indicating that the overall shape complementarity was a key factor in improving stabilization of the complex” in the discussion section to address the final comment by the reviewer. Overall, we would refrain from generalizing regarding the difficulty of introducing polar contacts. We would rather conclude that overall shape complementarity was more important than an additional polar contract between the ligand and the protein.

Reviewer #2 (Remarks to the Author)

I thank the authors for fully addressing all my concerns. It is exciting to hear that this publication will be backed up by a followup paper in due course!

I recommend publication of this manuscript.

Response: We thank reviewer 2 for supporting the publication of our work in its current form.

Reviewer #3 (Remarks to the Author)

The revisions provided by the authors have satisfied my critiques & comments. Specifically, I found the added information regarding the use of AnchorQuery informative from the perspective of a medicinal chemist who would want to learn how to apply the author's strategy. Further, clarifications, with more specific references towards prior works, enabled me to have a much better understanding of how the authors chose the compounds which were disclosed. I also found the further clarification within the body of the text which explained data which suggests molecular glue vs. protein binding to enhance my understanding of the authors data interpretation. Considering the characterization data provided within the supplemental document, for library compounds, I think it is quite clear that the compounds proposed match the presented characterization data. I would still strongly encourage the authors to add to their supplemental stating a reason why MS data is not provided on the per compound basis. As this paper is eventually indexed by search engines and read by scientists potentially out of context or without the MS expertise possessed by the authors, I believe it to be good practice to include a reason when certain unobtainable data is not included.

The current version of the manuscript meets my requirements for publication.

Response: We thank reviewer 3 for the positive feedback of the revised manuscript. In the SI, we provided additional MS and NMR data, following an editorial request. More specifically:

- For the key compounds **9** and **25**, we have acquired complete ^1H and ^{13}C NMR spectra as well as LC-MS data. These are now included in the revised SI (pages 28–30).
- For compound **31**, we have obtained ^1H NMR and LC-MS data. Due to instability upon prolonged standing, ^{13}C NMR spectra could not be reliably obtained.
- For **compounds 35, 36** and **37**, we were only able to collect ^1H NMR spectra, which are included despite being somewhat impure. These compounds bear highly reactive electrophilic warhead functionalities,

which we believe contribute to their notable instability, especially during purification and storage. Despite repeated attempts, isolation in sufficient purity and quantity for full characterization was not feasible. LC-MS data demonstrated instability as well. We already noted in the manuscript that these analogs were unstable. These compounds were included to illustrate structural diversity within the scope of our methodology but are not central to the conclusions of the manuscript, as unstable, inactive compounds. We have added a note in the SI to clarify these limitations and marked compounds 35-37 as partially characterized. We hope these efforts and explanations adequately address your concerns.